# Implicit motor adaptation patterns in a redundant motor task manipulating a stick with both hands

**Toshiki Kobayashi[1,2], Daichi Nozaki[1]\***

[1]Graduate School of Education, The University of Tokyo, Tokyo, Japan; [2]Japan Society for the Promotion of Science, Tokyo, Japan

## eLife Assessment

This study presents a **valuable** finding on how the sensorimotor control system deals with redundancy within our body, based on a novel bimanual task. The evidence supporting the authors' claims is **convincing**, as demonstrated over four different experiments. The work will be of interest to researchers from the motor control community and related fields, and further investigation into the interpretation of the findings could increase the generalisation of the study to a broader audience.

**Abstract** The remarkable ability of the motor system to adapt to novel environments has traditionally been investigated using kinematically non-redundant tasks, such as planar reaching movements. This limitation prevents the study of how the motor system achieves adaptation by altering the movement patterns of our redundant body. To address this issue, we developed a redundant motor task in which participants reached for targets with the tip of a virtual stick held with both hands. Despite the redundancy of the task, participants consistently employed a stereotypical strategy of flexibly changing the tilt angle of the stick depending on the direction of tip movement. Thus, this baseline relationship between tip-movement direction and stick-tilt angle constrained both the physical and visual movement patterns of the redundant system. Our task allowed us to systematically investigate how the motor system implicitly changed both the tip-movement direction and the stick-tilt angle in response to imposed visual perturbations. Both types of perturbations, whether directly affecting the task (tip-movement direction) or not (stick-tilt angle around the tip), drove adaptation, and the patterns of implicit adaptation were guided by the baseline relationship. Consequently, tip-movement adaptation was associated with changes in stick-tilt angle, and intriguingly, even seemingly ignorable stick-tilt perturbations significantly influenced tip-movement adaptation, leading to tip-movement direction errors. These findings provide a new understanding that the baseline relationship plays a crucial role not only in how the motor system controls movement of the redundant system, but also in how it implicitly adapts to modify movement patterns.

**\*For correspondence:**
nozaki@p.u-tokyo.ac.jp

**Competing interest:** The authors declare that no competing interests exist.

## Introduction

If a movement error occurs while reaching a target, the motor system automatically and implicitly updates the motor command for the next movement to reduce the error. This remarkable motor learning ability contributes not only to maintaining accurate movements in the presence of noise but also to flexibly adapting the movement to new environments. In the standard planar arm-reaching task, which has been widely used to study the mechanisms of motor learning, participants manipulate a visual cursor with their hands primarily by moving their shoulder and elbow joints (*Flash and Hogan, 1985*; *Gordon et al., 1994*; *Figure 1a*). The arm-reaching task has contributed to uncovering many

**Figure 1.** Kinematic redundancy and adaptation patterns. (**a**) In the ordinary planar arm-reaching task which primarily uses elbow and shoulder joints, there is no redundancy between the task goal ($x_c$, $y_c$) and the joint angles ($\theta_1$, $\theta_2$). (**b**) The movement pattern at each target is determined by the joint angles. (**c**) The adaptation patterns to a visual rotation are also determined. (**d**) This adapted joint angle pattern should be identical to that of voluntary reaching in the adapted direction. (**e, f**) In the case of planar reaching movement by a hypothetical limb with three joints ($\theta_1$, $\theta_2$, $\theta_3$), the cursor can reach the target in a large number of patterns. (**g**) Given that the motor system solves the task redundancy by optimization, we should observe a stereotypical pattern of joint angles according to the targets. (**h, i**) The question is whether, when a visual rotation is imposed on the cursor (end-effector relevant perturbation), the motor system adopts the same joint movement pattern as when voluntarily aiming in the adapted direction (*left*) or a new joint movement pattern (*right*). (**j**) The joint angles can be perturbed while the cursor position remains unchanged (end-effector irrelevant perturbation). (**k**) This perturbation can be ignored because it does not affect the cursor (*left*). Alternatively, the joint angles may be corrected (*right*).

important properties of motor adaptation when a novel environment is introduced (*Shadmehr and Mussa-Ivaldi, 1994*; *Thoroughman and Shadmehr, 1999*; *Mazzoni and Krakauer, 2006*; *Taylor and Ivry, 2011*; *Takiyama et al., 2015*; *Hayashi et al., 2016*).

Notably, there is no redundancy between hand position and joint angles in such a planar arm-reaching task. The joint angles are primarily determined by the target to be reached (*Figure 1b*). Because of this non-redundancy, when the motor system adapts the reaching movement to the visual rotation imposed on the cursor (*Figure 1c*, left), the adapted pattern of joint angles is also determined (*Figure 1c*, right). Naturally, the adapted pattern of joint angles should be identical to the pattern when reaching in the adapted direction (*Figure 1d*). However, our body systems for performing

practical motor tasks generally have more degrees of freedom than necessary to accomplish the tasks (*Bernstein et al., 1996*; *Singh et al., 2016*). For example, in the golf swing, there are countless patterns of whole-body configurations to achieve identical clubhead positions at ball contact.

The problem arising from such a redundant structure can be illustrated by considering the reaching movement of a hypothetical limb with three joints (*Figure 1e*). In this case, there are a large number of possible joint angle patterns for the same hand position (*Figure 1f*). However, given that the motor system solves task redundancy by minimizing motor cost (*Collins, 1995*; *O'Sullivan et al., 2009*) or movement error (*White and Diedrichsen, 2010*; *Zhang et al., 2018*), we should observe a stereotypical pattern of joint angles according to the targets (*Figure 1g*). The problem is how the motor system changes the direction of cursor movement to implicitly adapt to the visual rotation imposed on the cursor (*Figure 1h*). Are the joint angles after adaptation identical to those observed when aiming in the same adapted direction before adaptation? (*Figure 1i*, left). Alternatively, does the motor system adopt a new joint movement pattern? (*Figure 1i*, right). It has not been thoroughly investigated how the motor system coordinates to change the redundant body system to implicitly adapt to the perturbation to the cursor representing the position of the end-effector.

Redundancy poses another type of problem for implicit movement adaptation. If the joint angles are perturbed while the hand position remains unchanged (*Figure 1j*), how does the motor system respond to the perturbation? The motor system may simply ignore the perturbation because it does not affect the cursor position (*Figure 1k*, left). Alternatively, the motor system may attempt to correct the joint movements (*Figure 1k*, right). Conventionally, this type of problem has been investigated in the context of how the motor system responds to the movement variability that affects or does not affect task performance (i.e. task-relevant or task-irrelevant variability). Previous studies have consistently shown that the motor system actively corrects for task-relevant variability while allowing task-irrelevant variability (*Mosier et al., 2005*; *Latash, 2012*; *van Beers et al., 2013*; *Sternad, 2018*; *Scholz and Schöner, 1999*; *Scholz et al., 2003*). Optimal feedback control theory suggests that the strategies based on such a 'minimal intervention principle' are optimal in terms of achieving task goals (e.g. movement accuracy; *Todorov and Jordan, 2002*; *Todorov, 2004*; *Scott, 2004*; *Diedrichsen, 2007*; *Valero-Cuevas et al., 2009*; *Ronsse et al., 2010*; *Sternad et al., 2011*; *Selgrade and Chang, 2015*). These studies demonstrated that the motor system does not intervene movement change or variability as long as the task goal is achieved. Thus, the motor system should ignore the perturbations to joint angles that do not affect the task goal (*Figure 1k*, left). However, these studies did not directly test whether the redundant body movement pattern is also unchanged when the end-effector movement remains unchanged.

Conversely, it has also been reported that the motor system adapts to change the movement in response to the task-irrelevant perturbation (*Wolpert et al., 1995*; *Flanagan and Rao, 1995*; *Morehead et al., 2017*; *Diedrichsen et al., 2010*; *Schaefer et al., 2012*; *Franklin et al., 2016*; *Kim et al., 2019*). For example, when the cursor trajectory was curved only in the middle of the reaching movement, the hand movement was adaptively modified even though the perturbation did not cause a reaching error at the target (*Wolpert et al., 1995*; *Flanagan and Rao, 1995*). It should be noted that these previous studies created the task-irrelevant condition by applying the perturbation to the end-effector. In contrast, the present study focuses on how the motor system responds to the perturbation of the redundant body while the end-effector remains unaffected (*Figure 1i* and *Figure 1k*). In light of this, in this study, we use the terms 'end-effector relevant perturbation' (e.g. *Figure 1h*) and 'end-effector irrelevant perturbation' (e.g. *Figure 1j*) when appropriate, instead of the more commonly used terms 'task-relevant perturbation' and 'task-irrelevant perturbation'. The current study aims to develop a novel task that creates end-effector relevant and end-effector irrelevant perturbations.

To investigate the problem of how the motor system adapts the redundant body system to these perturbations, this study introduces a novel bimanual stick manipulation task specifically designed to address this issue (*Figure 2a*). Participants manipulated a virtual stick held with both hands to reach visual targets by the tip of the stick. Redundancy in the task is achieved by acknowledging that the same tip position can be achieved with different stick angles (*Figure 2b*). Importantly, this task allowed us to visually perturb the tip position (i.e. end-effector relevant perturbation) and the stick angle (i.e. end-effector irrelevant perturbation) either separately or simultaneously. Using this approach, we can directly investigate how the motor system makes use of the redundant body system when adapting to these two types of perturbations.

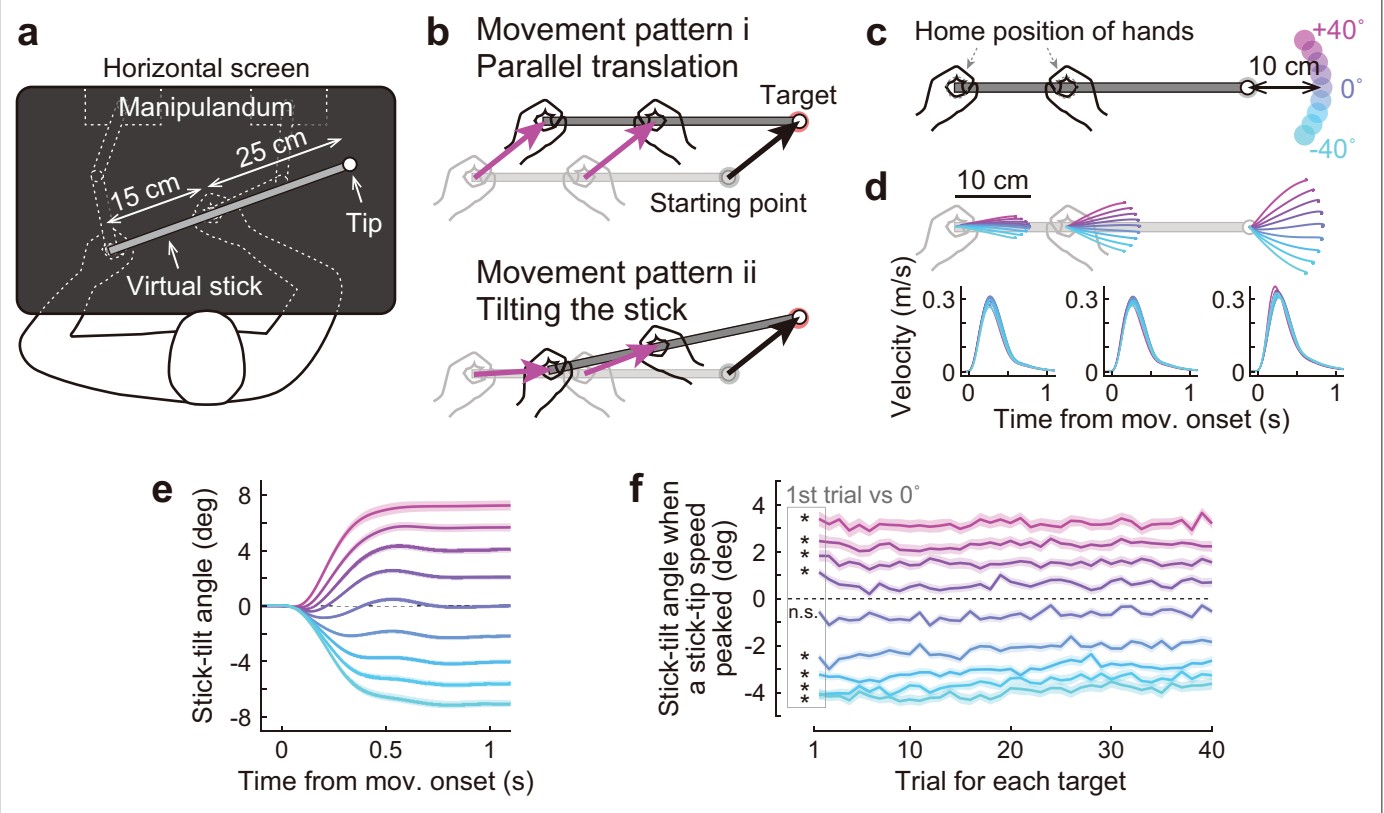

**Figure 2.** Bimanual stick manipulation task. (**a**) Participants held handles of a manipulandum with both hands and moved the right tip of a virtual stick on the monitor. (**b**) In this task, there are countless movement patterns to move the tip to the target. Two representative strategies are shown: parallel translation (*top*) and tilting the stick (*bottom*). (**c**) During the baseline phase, 72 participants were instructed to move the tip from a starting point to a target that appeared in one of nine directions. (**d**) Movement trajectories (*top*) and velocity profiles (*bottom*) of the left hand, right hand, and tip averaged across participants. Each color corresponds to the target direction shown in **c**. The shaded areas represent the SEM across participants. (**e**) The time-course of the stick-tilt angle shows that participants moved the tip while tilting the stick. The shaded areas indicate the SEM. (**f**) Trial-to-trial changes in stick-tilt angle at the peak velocity of the tip. The strategy of tilting the stick was observed from the beginning of the experiment (*t*-test compared with zero, |*t*(71)|>6.4, Bonferroni corrected p<0.001). The shaded areas indicate the SEM.

## Results

### Movement strategy to manipulate the stick

A total of 72 healthy participants took part in the study (Experiments 1–3). They were asked to manipulate a virtual stick with both hands (length = 40 cm; distance between hands = 15 cm) to move the right tip of the stick towards the targets (*Figure 2a*). This task is redundant because the same target can be reached in countless movement patterns. The participants could move both hands in parallel without tilting the stick (*Figure 2b*, top). Alternatively, they could perform the task by tilting the stick (*Figure 2b*, bottom). During the first part of the task (baseline phase: 360 trials) in all the experiments (Experiments 1–3), the participants moved the tip from a starting point to a target (movement distance = 10 cm) that appeared in one of nine directions (0°: horizontal direction,±10°,±20°,±30°, and ±40°; *Figure 2c*).

The trajectories of the tip and both hands appeared to be almost straight, but both hands were not moved in parallel, except when aiming at the 0° target (*Figure 2d*, top). The velocity profiles of the hands and tip were bell-shaped (*Figure 2d*, bottom), similar to a normal unimanual reaching movement (*Flash and Hogan, 1985*; *Gordon et al., 1994*). The time-course changes in the stick-tilt angle relative to the initial horizontal tilt angle showed that the participants moved the tip while slightly tilting the stick, except when aiming at the 0° target (*Figure 2e*). This tendency was observed from the beginning of the experiment (*t*-test compared with 0°, |*t*(71)|>6.4, Bonferroni corrected p<0.001; *Figure 2f*).

Movement strategies to manipulate the stick for individuals are illustrated on the plane expressed by two variables: tip-movement direction relative to a horizontal direction (TMD) and the stick-tilt angle relative to the initial horizontal state (STA; *Figure 3a*). Although each participant could have performed the task differently (*Figure 2b*), all participants showed stereotypical movement patterns: The stick-tilt angle was significantly modulated by the tip-movement direction (*t*-test compared with 0°, $|t(71)|>5.5$, Bonferroni corrected p<0.01; *Figure 3a*, inset). This monotonically increasing curve, which will be referred to as the 'baseline TMD–STA relationship', reflects a constraint between the direction of tip movement and the angle of stick tilt. The relationship showed that clockwise (CW) tip movement direction was accompanied by CW stick-tilt, and counterclockwise (CCW) tip-movement direction was accompanied by CCW stick-tilt. This constraint of movement patterns was likely created by minimizing the movement distance of the hands. If both hands are moved in parallel, the total distance between the initial and final positions of the hands should be 20 cm (10 cm for each) regardless of the targets; however, the experimentally observed distance was reduced by tilting the stick (*Figure 3b*). The average distance was a near-minimal value (versus parallel translation: Cohen's d=1.914, versus theoretical minimum: Cohen's d=0.334; *Figure 3c*).

## Experiment 1: Motor adaptation pattern to the end-effector relevant perturbation

During the second part of the experiment (adaptation phase: 240 trials), the target appeared only in the 0° direction, and three types of visual perturbations were introduced. Most importantly, this task enabled the introduction of end-effector relevant perturbation (Experiment 1), or end-effector irrelevant perturbation (Experiment 2), or a combination of both types of perturbations (Experiment 3). In Experiment 1 A (N=19), the tip-movement direction was visually rotated in the CCW direction around the starting position (tip perturbation), whereas the stick-tilt angle remained unchanged (*Figure 4a*). Thus, this perturbation produced a positional dissociation between vision (i.e. the stick) and proprioception (i.e. the hands). The amount of tip directional rotation was gradually increased by 1° per a trial up to 30° so that the participants did not notice the presence of perturbation.

*Figure 4b* illustrates how this visual perturbation changes the states of stick movement in visual and physical spaces represented by mapping between the tip-movement direction and stick-tilt angle. The movement in the visual space was the one visible to the participants, whereas the movement in the physical space was the actual hand movement, which was invisible to them. On a plane representing visual movement patterns (*Figure 4b*, left), the visual perturbation to the tip can be expressed as a shift of the state along the x-axis. The error of the tip-movement direction in the visual space needs to be compensated by changing the movement of both hands. On a plane representing physical movement patterns (*Figure 4b*, right), this requirement can be expressed as a shift of the task goal in the direction opposite to the visual perturbation.

Owing to the redundant nature of this task, participants could adapt to change the visual tip movement using different strategies. For example, they could change the tip movement without changing the stick-tilt while keeping the strategy of moving both hands in parallel (*Figure 4c*, top). Alternatively, they might adapt by changing the stick tilt as if they were aiming at a –30° target (*Figure 4c*, bottom). These strategies can be expressed as a state shift towards the new task goal line along the x-axis or along the baseline TMD–STA relationship in the physical plane (*Figure 4d*).

The experimental results demonstrated that the participants adapted to the tip perturbation by tilting the stick (*Figure 5—figure supplement 1*). On the visual plane (*Figure 5a*), after the adaptation, the tip-movement direction remained at the baseline level (paired *t*-test, $t(18) = -1.320$, p=0.203), while the stick was tilted significantly more in the CW direction than when aiming at the 0° target in the baseline phase (paired *t*-test, $t(18) = 10.003$, p<0.01). Thus, the tip perturbation was compensated by adaptation, but the adaptation produced an abnormal visual stick-tilt. This result is likely to support the idea of the minimal intervention principle because participants ignored this visual stick-tilt which did not directly influence the task.

On the physical plane, adaptation appeared to progress along the baseline TMD–STA relationship throughout the adaptation phase (*Figure 5b* and *Figure 5c*). The implicitly adapted tip-movement direction and stick-tilt angle were similar to those when voluntarily aiming at the –30° target in the baseline phase. Although the difference was small but significant for the tip-movement direction (paired *t*-test, $t(18) = -10.053$, p<0.01), there was no significant difference in the stick-tilt angle (paired

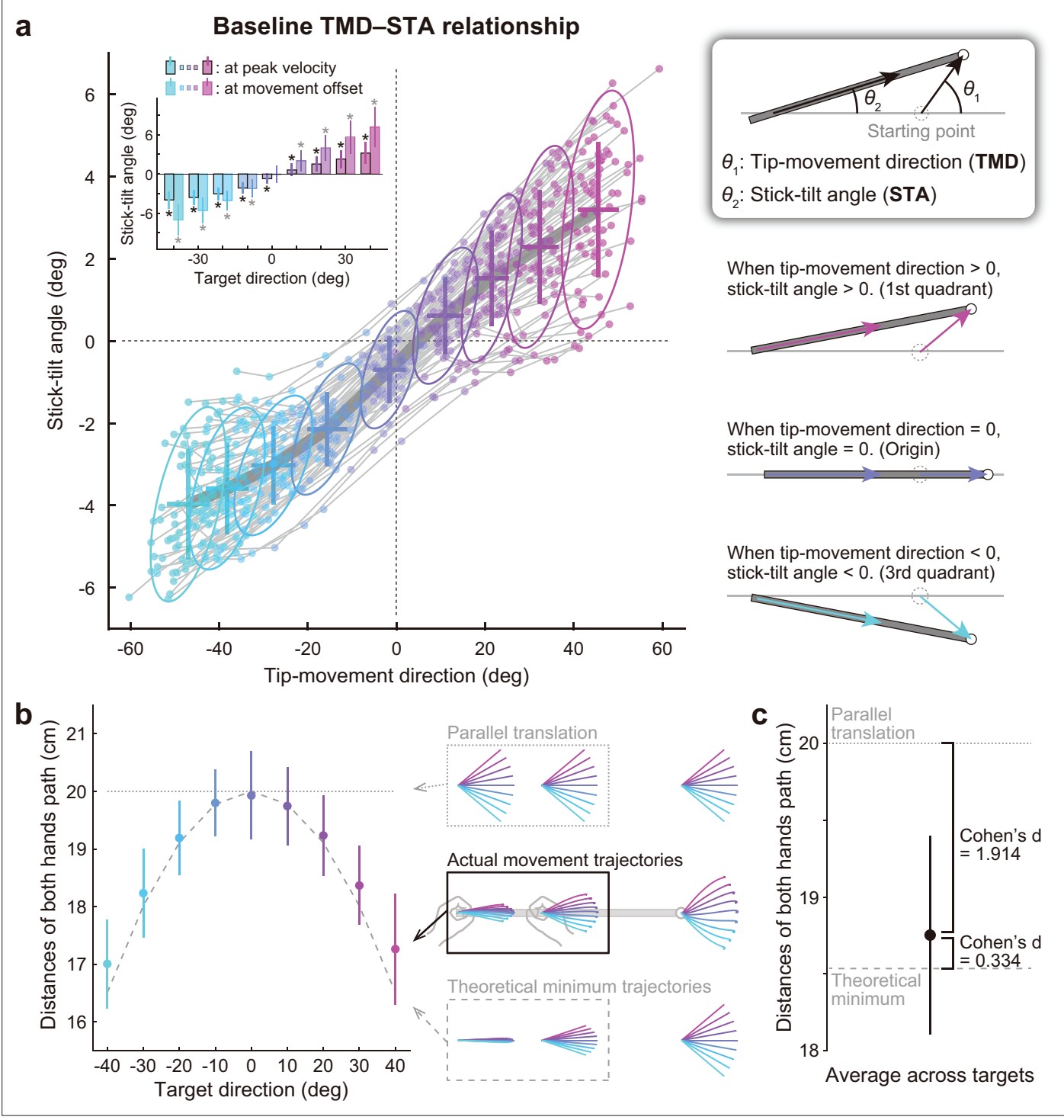

**Figure 3.** Relationship between the tip-movement direction and stick-tilt angle. (**a**) A relationship between each individual's tip-movement direction and stick-tilt angle, when the tip speed peaked is represented as the thin gray lines drawn with colored plots (N=72 participants). The averaged relationship is plotted as a bold gray line (baseline TMD–STA relationship) with the SD (crosses) and 95% confidence intervals (ellipses). The inset shows the stick-tilt angles at peak velocity and movement offset (error bars: SD, *: *t*-test compared with 0°, Bonferroni-corrected p<0.01). The monotonically increasing curve indicates the stereotypical strategy of manipulating the stick (diagrams on the right side). (**b**) Sum of the distances between the initial and final positions of both hands (i.e. the sum of the lengths of pink arrows in *Figure 2b*). The dotted and dashed gray lines indicate the distances when both hands are moved in parallel and the mathematically derived minimum distances by tilting the stick, respectively. The error bars denote the SD. (**c**) The distance averaged across target directions was closer to the minimum value (Cohen's d=1.914) than the parallel translation distance (Cohen's d=0.334). The error bar indicates the SD across participants.

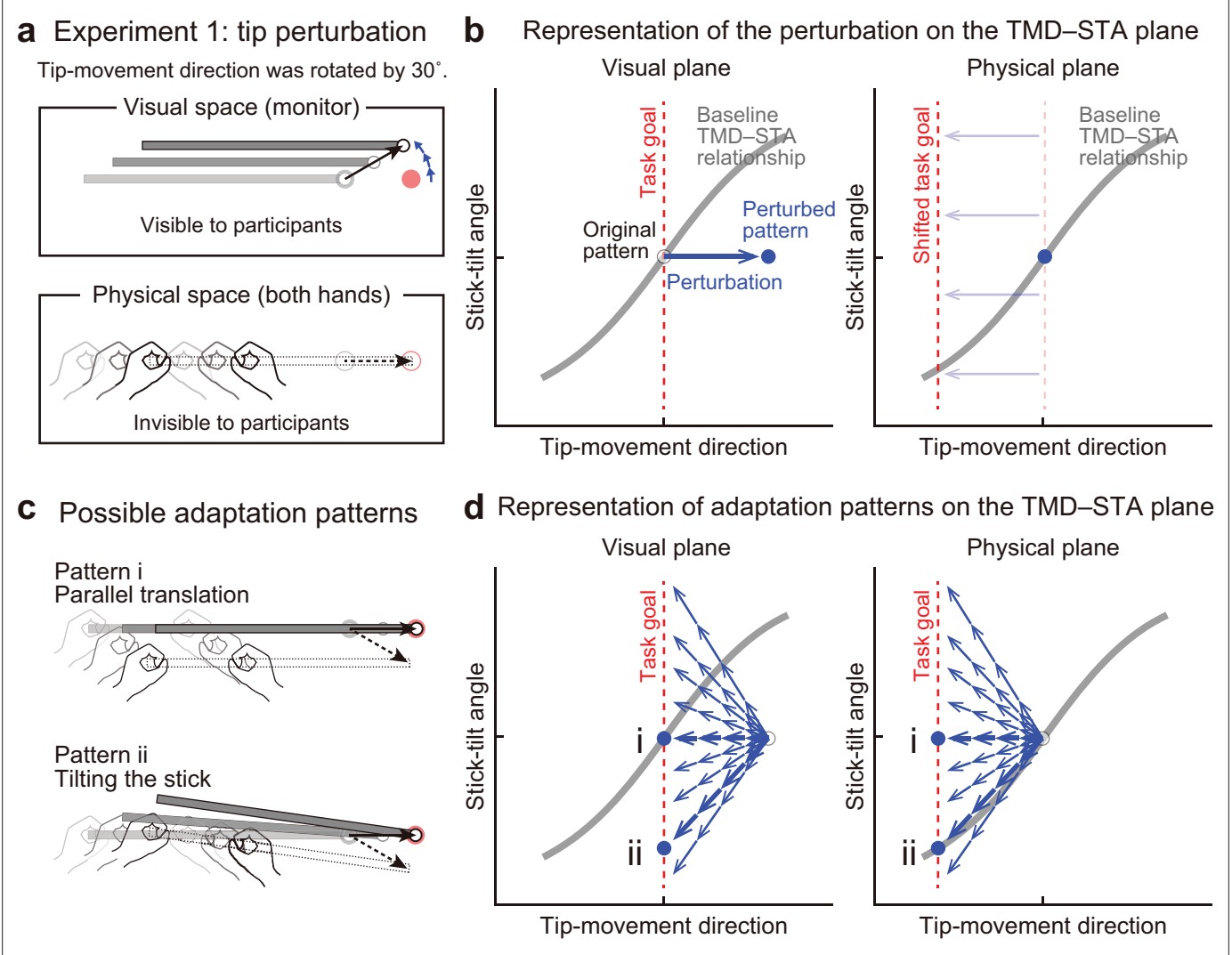

**Figure 4.** Experiment 1: End-effector relevant visual perturbations. (**a**) During the adaptation phase of Experiment 1, the tip-movement direction was rotated by 30° around the starting point (tip perturbation). The dotted and solid black arrows indicate the intended and actual tip trajectory, respectively. (**b**) The changes in the movement pattern are expressed as the state shift on the plane represented by mapping between tip-movement direction and stick-tilt angle (*left*: visual space, *right*: physical space). The dashed red lines denote the task goal (i.e. a target direction). On the visual plane, the perturbation (a blue arrow) shifts the movement pattern. On the physical plane, the task goal is shifted. (**c**) The participants could correct the tip error without (pattern i, *top*) or by (pattern ii, *bottom*) altering the stick-tilt angle. (**d**) The adaptive strategies are expressed as the shift on the visual plane (*left*) and the physical plane (*right*). The visual error caused by the perturbation could be compensated by different adaptation patterns (blue arrows): Representative patterns are expressed as bold arrows (patterns i and ii).

*t*-test, *t*(18) = −1.081, p=0.294). Thus, the motor system implicitly achieved adaptation to the tip perturbation as if it had voluntarily aimed at the −30° target (i.e. pattern ii in *Figure 4c* and *Figure 4d*; *Figure 5d*). This adaptation characteristic was also supported by the observation that the participants who tilted their stick more when aiming at the −30° target in the baseline phase exhibited a greater stick-tilt in the adapted movement patterns (R=0.601, p=0.007; *Figure 5e*). In Experiment 1B (N=13), we imposed the 30° tip perturbation abruptly to allow participants to use explicit strategy, and obtained nearly identical results (*Figure 5—figure supplement 2* and *Figure 5—figure supplement 3*).

As described above, the adapted tip-movement direction on the physical plane was significantly different from the tip-movement direction when aiming at the −30° target in the baseline phase (*Figure 5b*). In contrast, such significant difference was not observed for Experiment 1B (*Figure 5— figure supplement 2*). Why was such a difference only in Experiment 1 A? We speculated that this

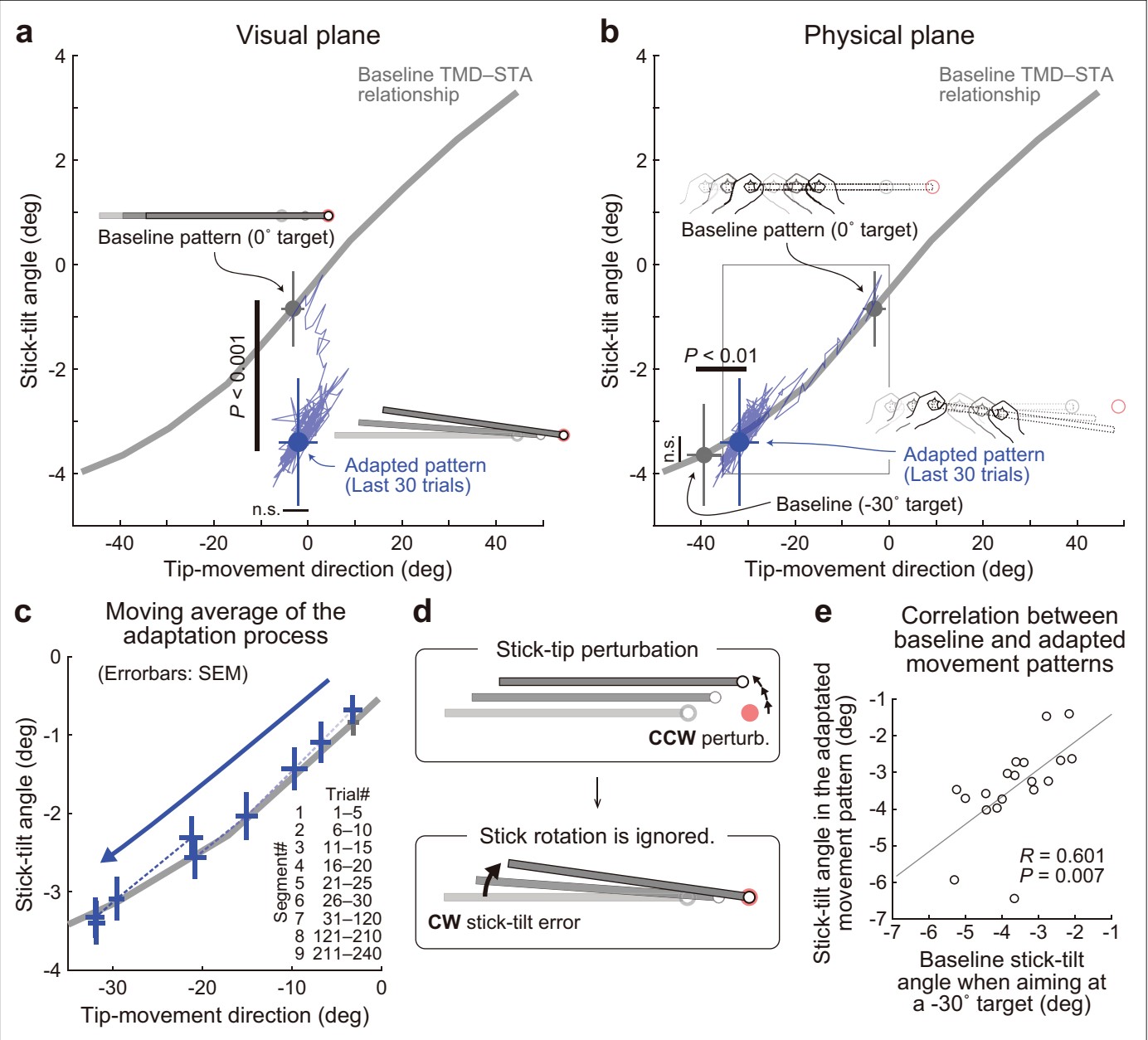

**Figure 5.** Experiment 1: Adaptation patterns to gradually imposed end-effector relevant perturbations. (**a**) In Experiment1A (N=19 participants), the adaptation trajectory (a thin blue line) on the visual plane gradually deviated from a baseline pattern (a gray circle) mainly along the y-axis. The tip-movement direction remained at the baseline level (paired *t*-test, *t*(18) = −1.320, p=0.203). In contrast, the stick-tilt angle after the adaptation (a blue circle) significantly differed from that in the baseline pattern when aiming at the 0° target (a gray circle; paired *t*-tests, *t*(18) = 10.003, p<0.001), indicating that the participants saw the abnormal visual stick-tilt. The plots indicate mean ± SD across participants. (**b**) On the physical plane, the adaptation trajectory followed the baseline TMD–STA relationship. Although the difference was small but significant for the tip-movement direction compared with the baseline pattern when aiming at the −30° target (paired *t*-test, *t*(18) = −10.053, p<0.01), there was no significant difference in the stick-tilt angle (paired *t*-test, *t*(18) = −1.081, p=0.294). (**c**) The area indicated by the squared outline in **b** is enlarged. The adaptive process can be divided into nine trial segments (trials 1–5, 6–10, 11–15, 16–20, 21–25, 26–30, 30–120, 121–210, and 211–240). (**d**) The motor system implicitly achieved adaptation to the tip perturbation as if it had voluntarily aimed at the −30° target. (**e**) Participants who showed a larger stick-tilt when aiming at the −30° target during the baseline phase also exhibited a larger stick-tilt in the adapted pattern (*R*=0.601, p<0.007).

The online version of this article includes the following figure supplement(s) for figure 5:

**Figure supplement 1.** Time-course of the movement pattern in the adaptation phase.

**Figure supplement 2.** Adaptation patterns to abruptly imposed end-effector relevant perturbations.

**Figure supplement 3.** Time-course of the movement pattern in the adaptation phase.

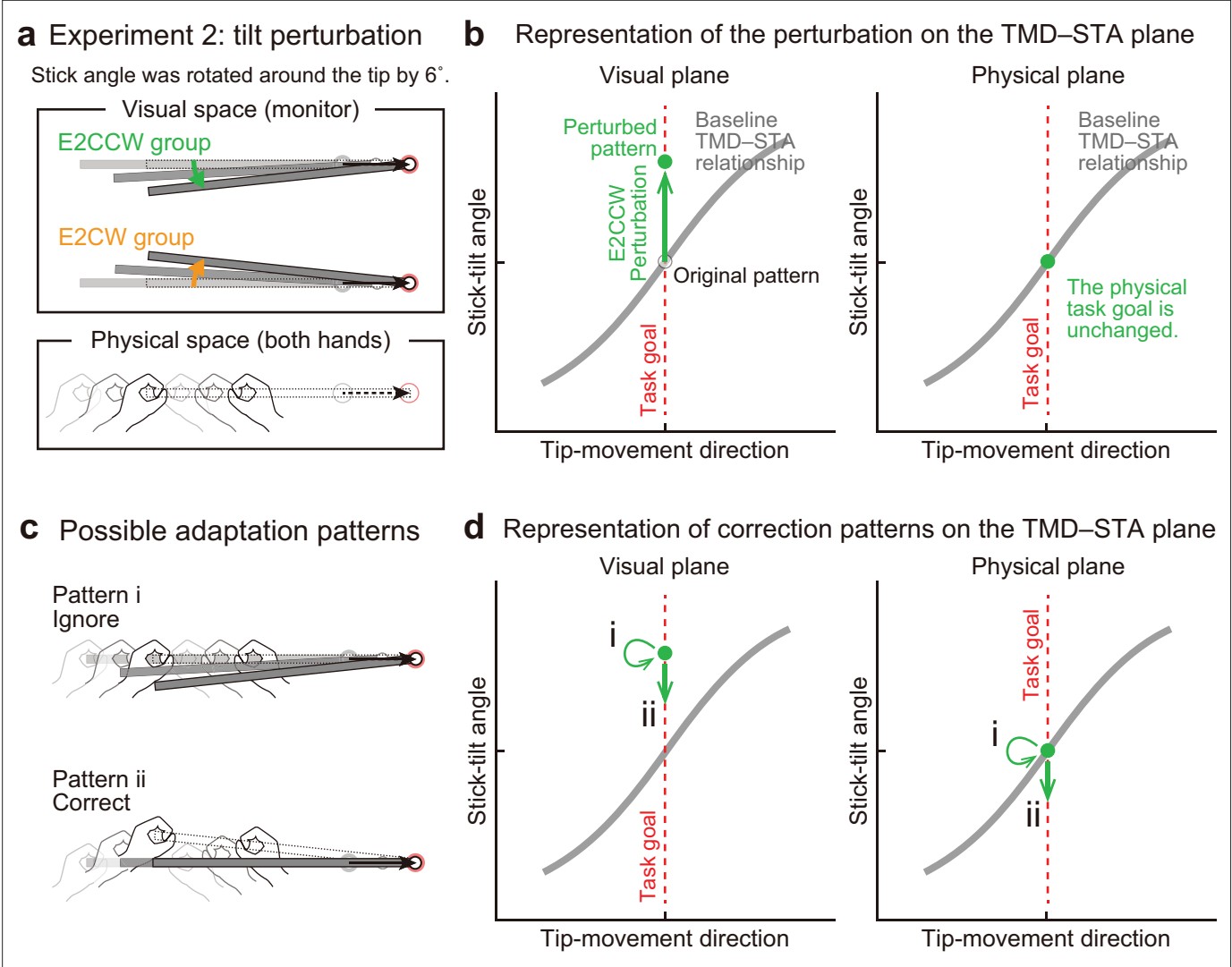

**Figure 6.** Experiment 2: End-effector irrelevant visual perturbations. (**a**) During the adaptation phase of Experiment 2, the stick-tilt angle was rotated around the tip. The perturbation of ±6° was imposed from the first trial. Participants received either the counterclockwise perturbation (green, E2CCW group) or the clockwise perturbation (orange, E2CW group). (**b**) On the visual plane, the perturbation is expressed as a vertical shift of the state along the task goal (dashed red line). On the physical plane, the perturbation does not influence the task goal. Note that, in both planes, the states are on the task goal lines. The formats are the same as in *Figure 4b*. (**c**) Participants may not change the movement pattern because the perturbation does not affect the tip position (pattern i, *top*). Alternatively, the perturbation can lead to a stick-tilt correction (pattern ii, *bottom*). (**d**) These strategies are expressed as a shift on the plane. Note that pattern i means that there is no change in the movement pattern.

difference was caused by the implicit adaptation to correct the abnormal visual stick-tilt angle. As shown in *Figure 5a*, after the adaptation, the stick was visually tilted in the CW direction. If the physical movement pattern is constrained by the baseline TMD–STA relationship, the CCW correction of the stick-tilt angle should be accompanied by the CCW change in tip-movement, which would interfere with the adaptation to the tip perturbation.

## Experiment 2: Motor adaptation pattern to end-effector irrelevant perturbation

Thus, it is still possible that the motor system does not necessarily ignore,but corrects for, the stick-tilt error that does not directly influence the tip position. Experiment 2 (N=20) was designed to directly test this possibility by imposing a stick-tilt perturbation around the right tip (i.e., end-effector irrelevant perturbation) (*Figure 6a*). As the tip moved from the starting point, the stick started to tilt around the tip up to ±6° when the tip reached the target (the tilt amount was proportional to the tip's

distance from the starting point). The ±6° tilt perturbation was chosen according to the stick tilt at the movement offset when reaching the ±30° target for the baseline phase (see *Figure 3a*). Ten participants participated in the CCW stick-tilt perturbation condition (E2CCW group: 6°), and the remaining ten participants took part in the CW stick-tilt perturbation condition (E2CW group: –6°). Because the amount was small, the participants did not notice the presence of perturbation throughout the experiment. Since this perturbation changed only the stick-tilt angle, it is expressed as a state shift along the y-axis on the visual plane (*Figure 6b*). According to the minimal intervention principle, the motor system does not need to change anything because the stick-tilt perturbation does not affect the tip position (i.e. the task goal; *Figure 6c* and *Figure 6d*, pattern i). However, if the motor system is actively compensating for the stick-tilt perturbation, it should change the stick-tilt angle (*Figure 6c* and *Figure 6d*, pattern ii).

Interestingly, the participants corrected the stick-tilt angle (*Figure 7—figure supplement 1*). On the visual plane, the stick-tilt angles remained different from those in the baseline phase (*Figure 7a*; two-way (direction of stick-tilt perturbation [i.e. groups], trial phase [first 30 trials or last 30 trials])) repeated-measures ANOVA, a significant main effect of groups: $F(1,18) = 54.457$, $p<0.001$; no effect of trial phase: $F(1,18) = 2.554$, $p=0.127$; no interaction: $F(1,18) = 1.613$, $p=0.220$. Thus, the adaptive changes did not fully compensate for the perturbation-induced stick-tilt errors. However, the data on the physical plane clearly demonstrated that participants implicitly changed the stick-tilt angle in the direction opposite to that of the perturbation from the early adaptation phase (*Figure 7b*): Without the correction, the data of the E2CCW and E2CW groups should have largely overlapped. However, with respect to stick-tilt angle, two-way repeated-measures ANOVAs revealed the main effect of stick-tilt perturbation direction ($F(1, 18)=12.483$, $p<0.01$) but not the main effect of trial phase ($F(1, 18)=3.331$, $p=0.085$). The interaction between these factors was not significant ($F(1, 18)=1.362$, $p=0.258$).

Unexpectedly, this stick-tilt correction was accompanied by undesirable tip-movement directional errors (*Figure 7c*). With respect to the tip-movement direction, two-way repeated-measures ANOVAs revealed the main effect of the direction of stick-tilt perturbation ($F(1, 18)=5.104$, $p=0.037$) but not the main effect of trial phase ($F(1, 18)=1.288$, $p=0.271$) (the interaction between these factors was not significant: $F(1, 18)=0.804$, $p=0.382$). Notably, these seemingly irrational adaptation patterns were consistent with the patterns predicted from the baseline physical TMD–STA relationship: CCW (CW) stick-tilt adaptation to the CW (CCW) stick-tilt perturbation slightly shifted the tip-movement direction in the CCW (CW) direction (*Figure 7d*).

We also examined how the amount of adaptation depended on the amount of stick-tilt when voluntarily aiming at the ±30° target in the baseline phase. The 6° stick-tilt perturbation should be insignificant for participants who tilt the stick at a larger angle; thus, this perturbation tends to be ignored. Conversely, for participants who did not tilt the stick, a 6° stick-tilt perturbation should be significant and must be corrected. *Figure 7e and f* show how the amount of stick-tilt correction (*Figure 7e*) and tip-movement directional change (*Figure 7f*) depended on the amount of stick-tilt angle when aim at the ±30° target in the baseline phase. As predicted, we observed a negative correlation (stick-tilt correction: $R=–0.572$, $p<0.01$; tip-movement directional change: $R=–0.440$, $p=0.052$), although the correlation for the tip-movement direction change was not significant.

Taken together, the results of Experiment 1 and Experiment 2 suggest that the implicit adaptation in the redundant stick-manipulating task is achieved by the following two mechanisms: First, the motor system attempts to correct for any perturbation in visual feedback, regardless of whether it is end-effector relevant or end-effector irrelevant. Second, the physical pattern of movement correction is constrained by the baseline TMD–STA relationship.

## Experiment 3: Interaction between end-effector relevant and end-effector irrelevant perturbations

Experiment 3 (N=20) was designed to further investigate the interaction between adaptations to end-effector relevant and end-effector irrelevant perturbations by applying both simultaneously (*Figure 8a*). We hypothesized that adaptation to the tip perturbation would be influenced by the direction of the simultaneously applied stick-tilt perturbation. Specifically, the CCW tip perturbation was combined with the CCW stick-tilt perturbation (E3CCW group: N=10) or with the CW stick-tilt perturbation (E3CW group: N=10). These visual perturbation patterns can be expressed as state shifts in the visual plane (green and orange arrows for E3CCW and E3CW groups, respectively, in

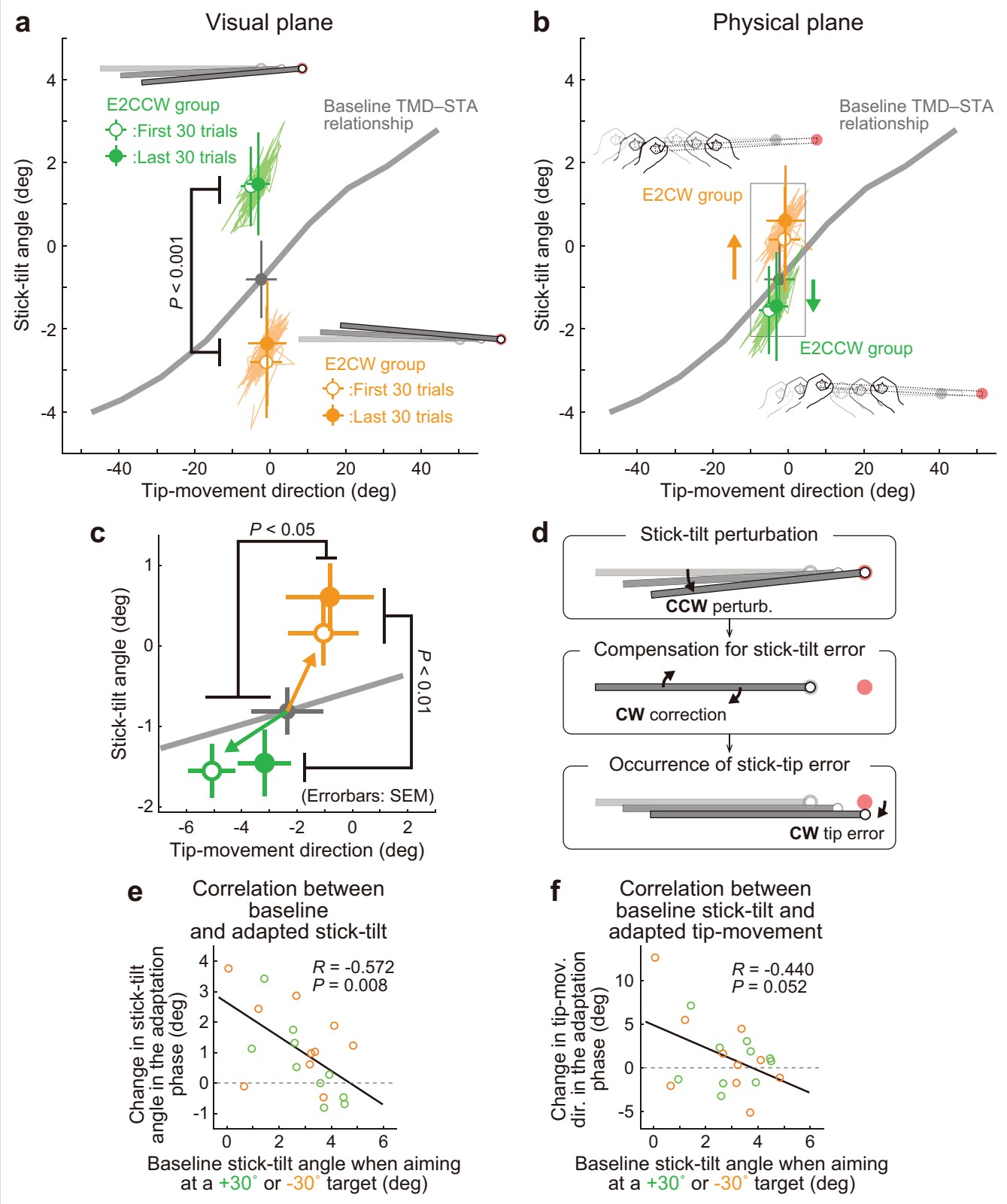

**Figure 7.** Experiment 2: Adaptation patterns to end-effector irrelevant visual perturbations. (**a**) Movement patterns are shown by green (E2CCW group) and orange (E2CW group) plots on the visual plane (N=10 participants for each group). The open and filled plots represent the movement patterns in the first and last 30 trials, respectively, during the adaptation phase (mean ± SD, across participants). (**b**) On the physical plane, the separated movement patterns between both groups showed that participants corrected the end-effector irrelevant stick-tilt errors. (**c**) The enlarged plot of the squared area in

*Figure 7 continued on next page*

*Figure 7 continued*

**b** shows that end-effector relevant tip-movement directional errors accompanied the adaptation. These effects persisted until the end of the adaptation phase (two-way repeated-measures ANOVAs). (**d**) Schematics of how the stick-tilt perturbation altered the movement patterns. (**e, f**) How participants tilted the stick during the baseline phase was negatively correlated with the stick-tilt angle and the tip-movement direction. To control for the individual variability, the baseline stick-tilt angle or tip-movement direction when aiming at the 0° target was subtracted from each value.

The online version of this article includes the following figure supplement(s) for figure 7:

**Figure supplement 1.** Time-course of the movement pattern in the adaptation phase.

*Figure 8b*). On the physical plane, the correction pattern required for the E3CCW group (*Figure 8c*, top) is aligned with the baseline TMD–STA relationship (green arrow in *Figure 8d*), from which we can predict that adaptation should proceed smoothly. In contrast, the correction pattern required for the E3CW group (*Figure 8c*, bottom) is orthogonal to the baseline TMD–STA relationship (orange arrow in *Figure 8d*), which may prevent smooth adaptation. We tested these predictions by simultaneously

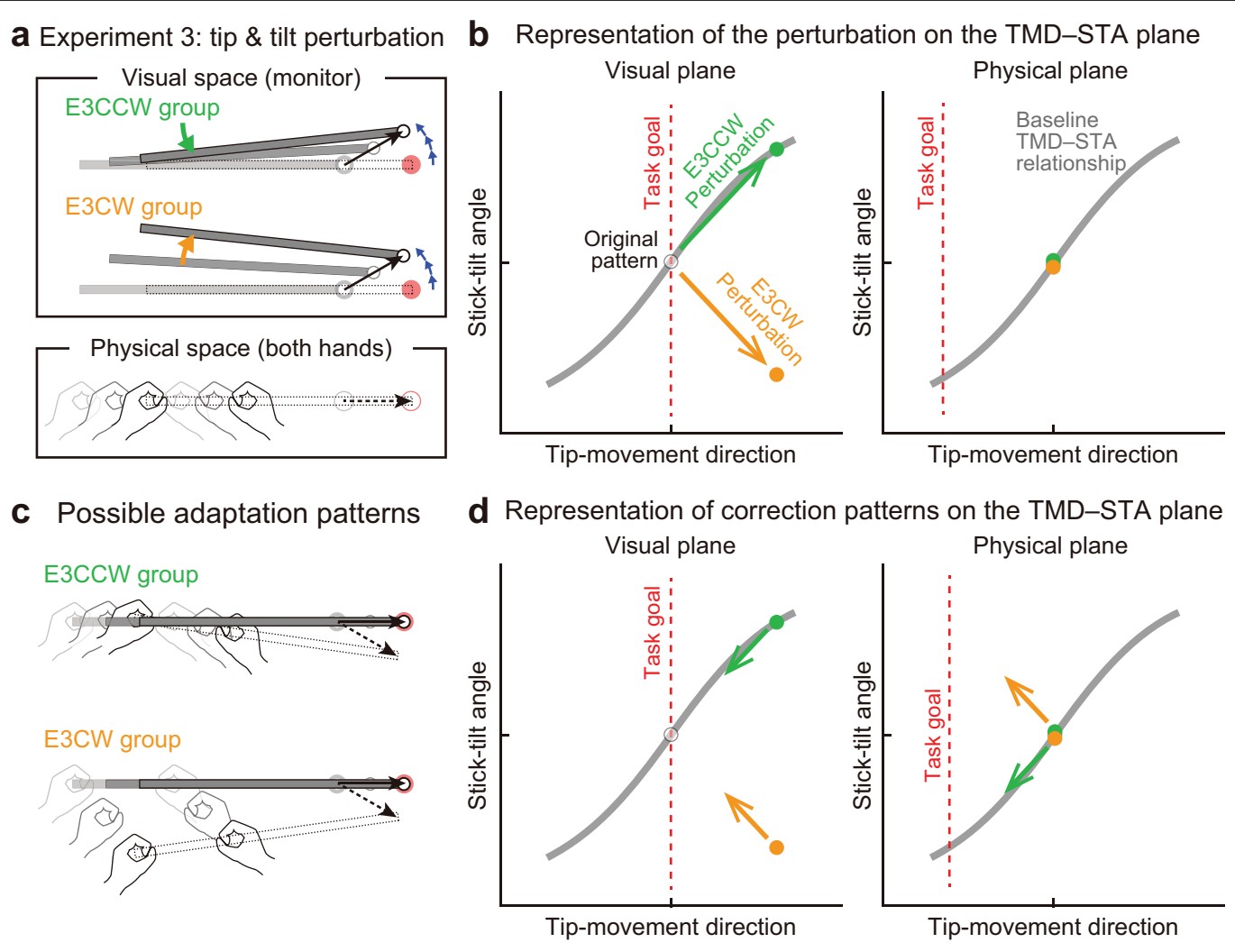

**Figure 8.** Experiment 3: Interaction between end-effector relevant and end-effector irrelevant visual perturbations. (**a**) During the adaptation phase of Experiment 3, both the CCW tip perturbation and the CCW or CW stick-tilt perturbation were introduced. The participants were assigned to the CCW stick-tilt rotation condition (green, E3CCW group) or the CW stick-tilt rotation condition (orange, E3CW group). (**b**) These visual perturbations can be expressed as state shifts on the visual plane (green and orange arrows for E3CCW and E3CW groups, respectively). (**c**) The adaptation pattern to reduce visual errors (*top*: E3CCW group, *bottom*: E3CW group). (**d**) On the physical plane, the correction pattern required for the E3CCW group is aligned with the baseline TMD–STA relationship (green arrow), whereas the correction pattern required for the E3CW group is orthogonal to the baseline TMD–STA relationship (orange arrow).

imposing the tip perturbation which was the same as the perturbation used in Experiment 1 A and the stick-tilt perturbation which was the same as the perturbation used in Experiment 2. Again, the participants were unaware of the perturbations throughout the experiment.

Contrary to the predictions, the difference in stick-tilt perturbation directions did not affect the final adapted movement patterns (*Figure 9—figure supplement 1*): As shown in the visual plane (*Figure 9a*), the difference in stick-tilt angle remained unchanged (paired *t*-test, $t(18) = 6.920$, p<0.001). Similarly, in the physical plane (*Figure 9b*), there were no significant differences in movement pattern between the two groups (tip-movement direction: paired *t*-test, $t(18) = -0.479$, p=0.638, stick-tilt angle: paired *t*-test, $t(18) = -0.386$, p=0.704). However, a trial-by-trial variability of the movement pattern in the latter adaptation phase (trial# 121–240) showed a slight difference between groups (*t*-test, tip-movement direction: $t(18) = -2.046$, p=0.056, stick-tilt angle, $t(18) = -3.235$, p=0.005), suggesting that the adaptation processes were implicitly affected by combination of perturbations (*Figure 9a*, inset). In partial agreement with our predictions, two groups showed different adaptation processes: trial-dependent trajectories to the final adapted pattern were clearly separated for both groups (*Figure 9c*). Indeed, a repeated-measures multivariate analysis of variance (MANOVA) showed a significant effect of group (Pillai's Trace $F(8, 162)=0.155$, p<0.001) and trial segment (Pillai's Trace $F(8, 162)=0.719$, p<0.001); however, no significant interaction was observed (Pillai's Trace $F(8, 162)=0.039$, p=0.982). As a result, the adaptation of the tip movement was more delayed in the E3CW group (trial segment 2: *t*-test, $t(18) = -3.684$, Bonferroni-corrected p<0.05; *Figure 9d*).

## Discussion

Our bodies have more degrees of freedom than necessary to achieve a relatively low-dimensional task goal. Understanding how the motor system resolves this redundancy remains a challenging problem (*Bernstein et al., 1996*). In this study, we investigated the problems of motor control and adaptation specific to kinematically redundant systems. The first problem is how the motor system coordinates the redundant body movements to control the end-effector (*Figure 1f*). The second problem is how the motor system coordinates to modify redundant body movements when implicitly adapting to the perturbation imposed on the end-effector (*Figure 1i*). The third problem is how the motor system responds to the perturbation to the redundant dimension that does not directly affect the end-effector (*Figure 1k*). These two types of perturbations have been conventionally referred to as 'task-relevant' and 'task-irrelevant' perturbations. However, it should be noted that previous studies investigating the effect of task-irrelevant perturbations have applied perturbations to the end-effector (e.g. cursor) without affecting task performance (*Wolpert et al., 1995*; *Flanagan and Rao, 1995*; *Diedrichsen et al., 2010*; *Schaefer et al., 2012*; *Franklin et al., 2016*; *Kim et al., 2019*). In contrast, the present study focused on investigating the effect of perturbations on the redundant body while leaving the end-effector unaffected (see *Figure 1k*). To clearly emphasize these differences, we specifically used the terms 'end-effector relevant perturbation' and 'end-effector irrelevant perturbation'.

To investigate these issues, we designed a novel stick manipulation reaching task. This redundant motor task enables the investigation of adaptation patterns in the redundant system following the introduction of perturbations that are either end-effector relevant, end-effector irrelevant, or both. Because our experiments were conducted with a specific configuration (stick length, hand distance, target location etc), we must be cautious about the generalizability of the results to other configurations, but we obtained the following four main results. First, we showed that participants adopted a stereotyped strategy to reach the target with the stick tip, and that the movement pattern (stick tip movement and stick tilt angle) was constrained by a baseline TMD–STA relationship. Second, when only end-effector relevant perturbations were imposed on the tip, the motor system attempted to modify the physical movement patterns along the baseline TMD–STA relationship to achieve the task goal (Experiment 1). Third, even when perturbations were applied only to the end-effector irrelevant dimension (i.e. the stick-tilt angle), the motor system still adapted to the end-effector irrelevant perturbation and this adaptation was accompanied by an end-effector relevant error (Experiment 2). Finally, when both types of perturbations were imposed simultaneously, the adaptation processes were influenced depending on the combination of perturbations (Experiment 3).

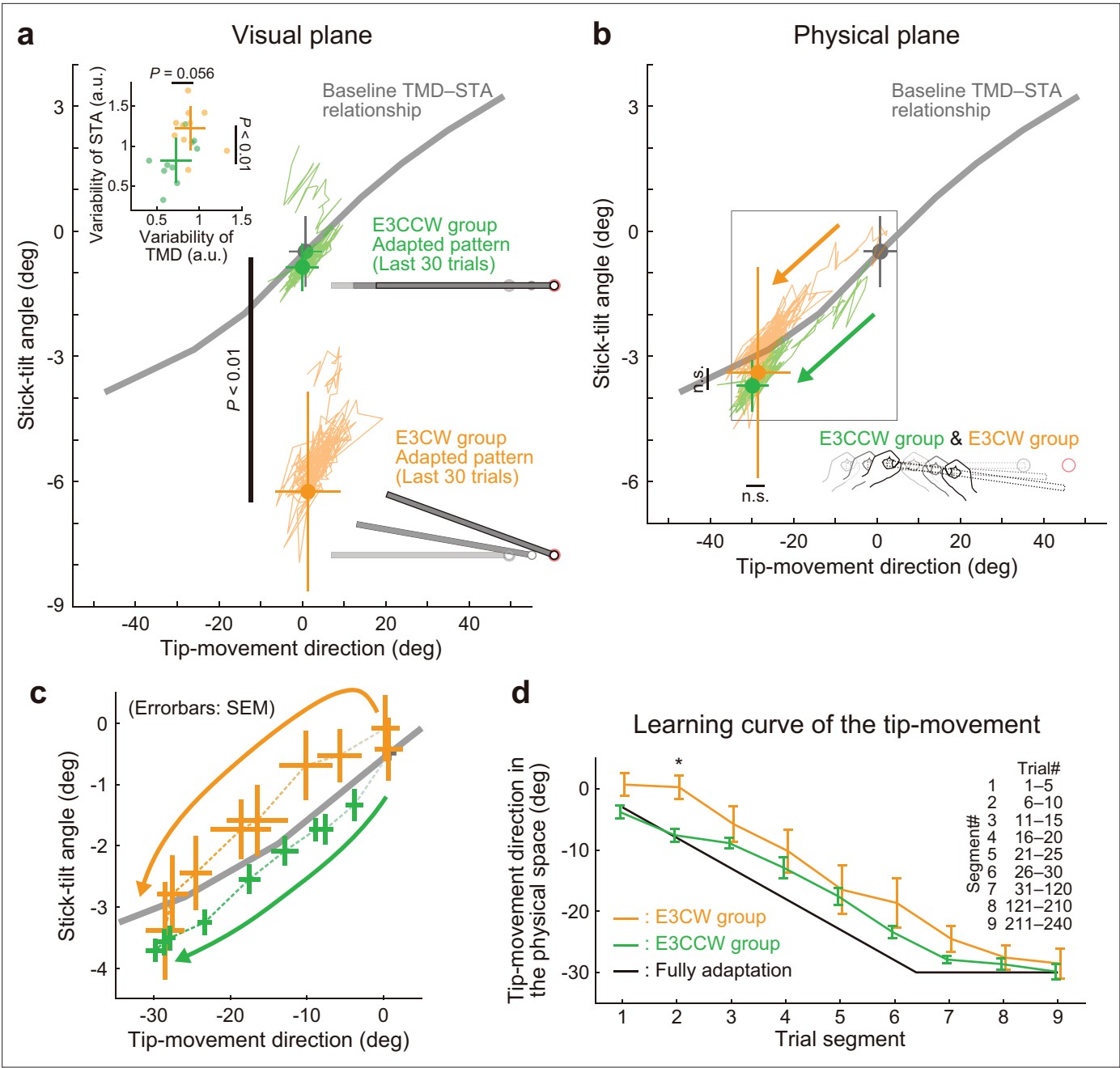

**Figure 9.** Experiment 3: Adaptation patterns to end-effector relevant and end-effector irrelevant visual perturbations. (**a**) Adaptation patterns on the visual plane are shown (same formats as *Figure 5a*). A green (orange) plot represents the adapted movement pattern in the E3CCW group (E3CW group) (N=10 participants for each group). The tip-movement direction did not significantly differ (*t*-test, *t*(18) = −0.473, p=0.642), while the stick-tilt angle remained a significant difference between groups (*t*-test, *t*(18) = 6.920, p<0.001). A trial-by-trial variability of the movement pattern in the latter adaptation phase (trial# 121–240) showed a difference between groups (*t*-test, tip-movement direction: *t*(18) = −2.046, p=0.056, stick-tilt angle, *t*(18) = −3.235, p=0.005) (**b**) Similarly, the adapted movement patterns on the physical plane did not show a significant difference (*t*-test, tip-movement direction: *t*(18) = −0.479, p=0.638, stick-tilt angle: *t*(18) = −0.386, p=0.704). (**c**) However, the adaptive processes appeared to be different. The movement patterns of the two groups are compared using a repeated-measures MANOVA. A multivariate analysis revealed a significant effect of group (Pillai's trace *F*(8, 162)=0.155, p<0.001) and trial segment (Pillai's trace *F*(8, 162)=0.719, p<0.001); no significant interaction can be observed (Pillai's trace *F*(8, 162)=0.039, p=0.982). The definition of trial segments is the same as that shown in *Figure 5c*. (**d**) The learning curves for the tip-movement. The adaptation in the E3CW group was always delayed but showed a significant difference only during the early phase (trial segment 2: *t*-test, *t*(18) = −3.684, Bonferroni-corrected p<0.05).

The online version of this article includes the following figure supplement(s) for figure 9:

**Figure supplement 1.** Time-course of the movement pattern in the adaptation phase.

## Kinematic relationship between end-effector relevant and end-effector irrelevant dimensions

In our stick manipulation task, participants were asked to reach the tip of the stick to the targets. Participants can achieve the reaching task in different ways. For example, they can use a simple strategy by moving both hands in the same direction (*Figure 2b*, top) or a more complicated strategy by tilting the stick (*Figure 2b*, bottom). Nevertheless, they showed stereotypical patterns of moving the stick tip by tilting the stick from the beginning of the task (*Figure 2f*): Reaching the CW (or CCW) targets was accompanied by the CW (or CCW) stick tilt (*Figure 3a*). Such stereotypical movement patterns have been reported for the movements of the joints in the redundant three-dimensional reaching movements (*Lacquaniti and Soechting, 1982*; *Proietti et al., 2017*). Notably, compared with the case where both hands moved in parallel, the movement distances of both hands to perform the task were significantly reduced to a near-optimal level (*Figure 3b* and *Figure 3c*), which contributed to the reduction of work required (*Collins, 1995*; *O'Sullivan et al., 2009*). As a result of this optimization process, tip-movement and stick-tilt were associated with each other by the baseline TMD–STA relationship (*Figure 3a*). Physically, this relationship constrains how the motor system should move the stick. Visually, it tells the motor system how the stick movement should be seen.

## The strength of the task to investigate the motor adaptation in redundant system

Our task has several advantages. First, the task has created redundancy in a natural way. Manipulating a stick with both hands is relatively common in everyday life (using a broom, golf or bat swing etc.). This familiarity with the task was reflected in *Figure 2f* which shows that all participants started tilting the stick to reach the tip toward the targets from the beginning of the experiment. This characteristic is ideal for investigating how this inherent movement pattern influences motor adaptation. Second, this task allows us to investigate how the perturbation to the tip-movement direction leads to a change in both the tip-movement direction itself and the stick-tilt angle. It has not been investigated how the motor system implicitly achieves adaptation to the end-effector perturbation (i.e. the end-effector relevant perturbation) by changing the movement patterns of the redundant body system. Third, this task can create a novel condition in which the perturbations are imposed on the redundant system while the tip position remains unchanged (i.e. the end-effector irrelevant perturbation). No previous studies have investigated how the motor system implicitly responds to such a perturbation by altering (or not altering) the movement of the redundant body system.

## Adaptation to end-effector relevant perturbation

When the end-effector relevant perturbation was implicitly imposed to the tip-movement direction (visual rotation by 30° CCW), the participants adapted to change the tip-movement direction in the opposite CW direction (Experiment 1), as reported in many previous studies using an ordinary reaching movement (*Cunningham, 1989*; *Krakauer et al., 2000*). Importantly, our task helped to investigate how such motor adaptations in the end-effector relevant dimension (tip-movement direction) are accompanied by the movement pattern changes in the end-effector irrelevant dimension (stick-tilt angle). Participants implicitly corrected the tip-movement direction by changing the stick tilt as if they were voluntarily aiming at the –30° target. This adaptation can be visualized as a state shift along the baseline TMD–STA relationship in the physical plane (*Figure 5b*). The baseline TMD–STA relationship originally contributes to dictate how the motor system voluntarily moves the stick. It is likely that the motor system also uses this relationship as a scaffold to implicitly guide how to correct the tip-movement direction. *Hayashi et al., 2016* have demonstrated that the adaptation of the standard reaching movement to a visual perturbation was significantly influenced by the baseline relationship between the movement and target directions (i.e. a visuomotor map). After distorting the visuomotor map so that the movement direction was more (or less) sensitively modulated by the target direction, the movement adaptation was enhanced (or reduced). The present finding that the baseline movement pattern is used for the implicit adaptation is consistent with this previous finding.

Notably, this adaptation pattern changed the visual feedback of the stick tilt. When reaching the 0° target in the baseline condition, the stick-tilt angle was almost 0° (*Figure 3a*). However, after adaptation, the amount of stick-tilt was equivalent to that when aiming at the –30° target (*Figure 5b*). Thus, the visually observed stick-tilt after adaptation was different from that observed in the baseline

condition (*Figure 5a*). Nevertheless, the adapted physical movement pattern persisted to follow the baseline TMD–STA relationship, suggesting that the motor system mainly paid attention to end-effector relevant tip-movement and ignored end-effector irrelevant stick-tilt angle. This adaptation pattern seems to be consistent with the minimal intervention principle.

However, a slight difference in the adaptation patterns between Experiment 1 A and Experiment 1B implies a violation of the minimal intervention principle. In Experiment 1 A, in which a 30° visual rotation was implicitly introduced by gradually increasing its amplitude, the final adapted tip-movement direction was slightly but significantly different from that when voluntarily aiming at the adapted direction (i.e. –30° target; *Figure 5b*). In contrast, such a residual difference was not observed in Experiment 1B, where participants could use an explicit aiming strategy (*Figure 5—figure supplement 2*). Visuomotor adaptation is often incomplete even after sufficient training, and residual errors are observed (*Krakauer et al., 2000*; *Ethier et al., 2008*; *Malone et al., 2011*; *Vaswani et al., 2015*). Although the presence of residual errors has been attributed to the imbalance between learning and forgetting, a recent study demonstrated that only the implicit learning system is responsible for producing residual errors by modulating error sensitivity with the consistency of errors in previous trials (*Albert et al., 2021*). We speculate that an alternative mechanism is involved in the generation of a residual difference in Experiment 1 A. If the motor system does not ignore the abnormal stick-tilt but attempts to correct it, and this correction pattern is physically constrained by the baseline TMD–STA relationship, the stick-tilt correction should accompany the de-adaptation of the tip-movement direction and lead to the generation of a residual error. As discussed below, the results of Experiment 2 and Experiment 3 support this idea.

## Adaptation to end-effector irrelevant perturbation

Our task can create a novel end-effector irrelevant stick-tilt perturbation and allows us to study how the motor system responds to it. According to the minimal intervention principle, the motor system should ignore stick-tilt perturbations because they do not influence task goal achievement (i.e. so called 'task-irrelevant' perturbation). However, inconsistent with this idea, Experiment 2 clearly demonstrated that the motor system responded to the stick-tilt perturbations (*Figure 7b* and *Figure 7c*). This result indicates that the motor system predicts the visual feedback of an end-effector irrelevant stick-tilt angle when performing the task and tries to eliminate the dissociation between the observed and predicted stick-tilt angle (i.e. sensory prediction error). This idea was corroborated by the fact that participants who demonstrated a shallower TMD–STA relationship during the baseline phase showed a greater amount of stick-tilt correction (*Figure 7e*). The ±6° stick-tilt perturbation should cause the larger prediction error for these participants.

One might consider that previous studies have already shown that the motor system responds to the task-irrelevant perturbations (*Wolpert et al., 1995*; *Flanagan and Rao, 1995*; *Diedrichsen et al., 2010*; *Schaefer et al., 2012*; *Franklin et al., 2016*; *Kim et al., 2019*). However, the task-irrelevant conditions implemented in these previous studies are quite different from the end-effector irrelevant condition in our study. These studies created the task-irrelevant condition by combining the perturbation of visual cursors with the manipulation of movement trajectory (*Wolpert et al., 1995*; *Flanagan and Rao, 1995*; *Morehead et al., 2017*), target/cursor shape (*Diedrichsen et al., 2010*; *Schaefer et al., 2012*; *Kim et al., 2019*), or applying the simultaneous equal perturbations to the target and cursor (*Franklin et al., 2016*). Therefore, these perturbations are not 'end-effector irrelevant'. Exceptionally, in several previous studies, an end-effector irrelevant perturbation was implemented in a bimanual postural task to investigate postural reflex activity (*Dimitriou et al., 2012*; *Omrani et al., 2013*). However, to our knowledge, such an end-effector irrelevant perturbation has never been used to investigate the motor adaptation pattern in a redundant system. Furthermore, as discussed in the next section, our task design revealed a novel aspect of motor adaptation to the end-effector irrelevant perturbation.

## Adaptation to end-effector irrelevant perturbation influences end-effector relevant performance

Unexpectedly, Experiment 2 shows that adaptation to the end-effector irrelevant stick-tilt perturbation was accompanied by an increase in the end-effector relevant tip-movement direction error (*Figure 7c*). This counterintuitive result can be explained by the constraint of the adaptation pattern along the

baseline TMD–STA relationship (*Figure 3a*). According to this constraint, CCW (or CW) stick-tilt adaptation should change the tip-movement direction in the CCW (or CW) direction (*Figure 7c*). The influence of stick-tilt perturbation on tip-movement direction error tended to be more pronounced for participants with a smaller stick-tilt angle at baseline, that is those with a shallower baseline TMD–STA relationship (*Figure 7f*), further supporting this idea. The results of Experiment 3 also support the existence of such an interaction between tip-movement and stick-tilt adaptation. When the tip-movement and stick-tilt perturbations were imposed simultaneously, the adaptation trajectories on the physical plane were significantly influenced by the perturbation combination patterns (*Figure 9c*). If the motor system had simply attempted to correct the stick-tilt to reduce the sensory prediction error, this spill-over effect on the tip-movement direction would not have been observed.

Given that the baseline TMD–STA relationship (*Figure 3a*) reflects the motor cost minimization, it is likely that the motor system is attempting to reduce both the sensor prediction error and the motor cost. One remaining question is why the motor system needs to correct for end-effector irrelevant stick-tilt errors in the first place. We speculate that there may be two reasons. First, a possible simple explanation is that the motor system predicts all sensory consequences generated by the action and attempts to correct any differences between the predicted and actual sensory information, even when the task is successfully achieved (*Mazzoni and Krakauer, 2006*; *Schaefer et al., 2012*; *Morehead et al., 2017*; *Kim et al., 2019*). Similarly, it is possible that the correction for end-effector irrelevant stick errors reflects the motor system's attempt to resolve the dissociation between stick (i.e. vision) and hands (i.e., proprioception) caused by the perturbation.

Second, given that tip-movement direction and stick-tilt angle are tightly coupled (i.e. the presence of baseline TMD–STA relationship), the presence of an end-effector irrelevant stick-tilt visual error implies the presence of an end-effector relevant tip-movement direction error. In other words, the motor system interprets end-effector irrelevant errors as evidence that end-effector relevant errors have been also produced. When sensory information contains noise and/or uncertainty, the reliability of sensory information estimation can be improved by utilizing additional sensory information (*Ernst*

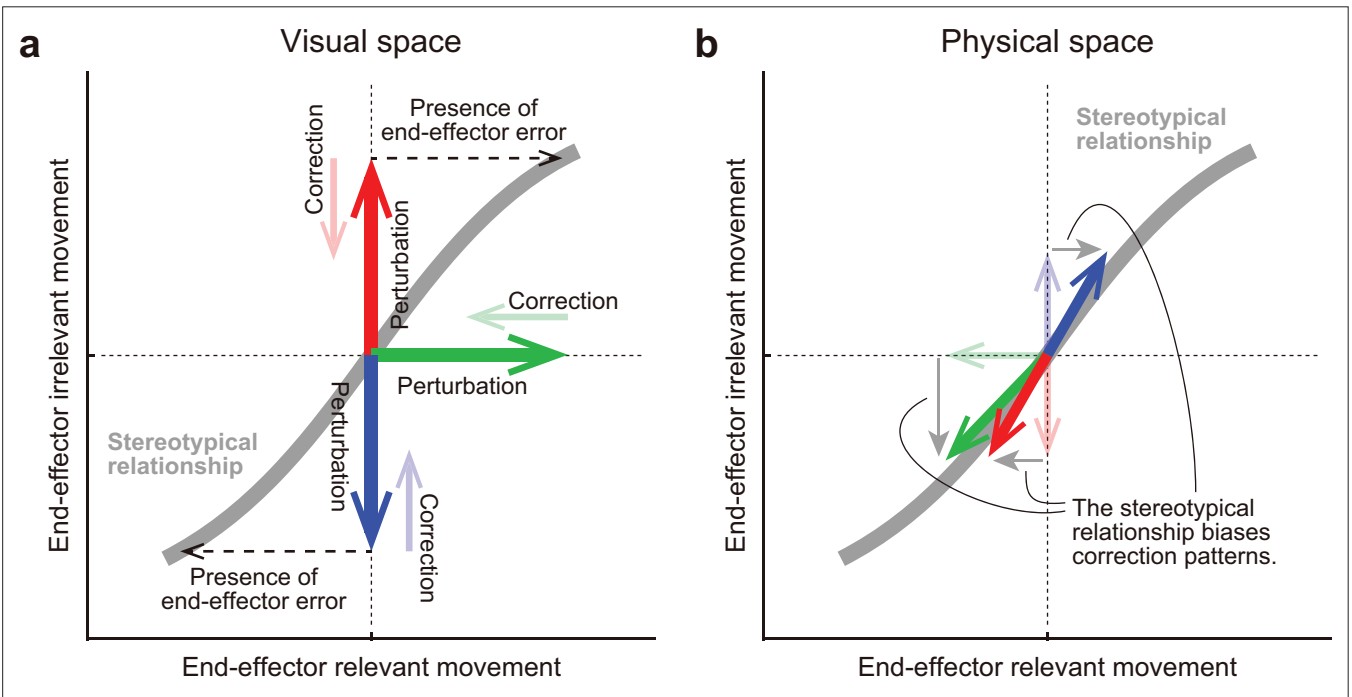

**Figure 10.** A hypothesis for how the motor system alters the movement pattern of redundant system. The motor system develops a stereotypical relationship in redundant space, possibly through an optimization process (a gray line). (**a**) In the visual space, this relationship plays a crucial role for the motor system in how it should implicitly respond to the various visual perturbations (bold arrows) in the redundant space. The motor system tries to eliminate the dissociation of the perturbed state from the original state (thin arrows). According to the relationship, the motor system interprets end-effector irrelevant errors (red or blue) as evidence that end-effector relevant errors have been also produced (black dashed arrows). (**b**) However, the correction patterns are constrained by the relationship in the physical space. The relationship determined which movement pattern the motor system should perform. Thus, the correction patterns gravitate toward the stereotypical relationship (bold arrows).

*and Banks, 2002*). From this viewpoint, utilizing stick-tilt information may be useful for improving the accuracy of the estimation of the tip-movement direction.

*Figure 10* illustrates the possible mechanism by which the motor system alters the movement pattern of the redundant system when encountering visual errors. The motor system develops a stereotyped movement pattern through an optimization process that is expressed by abaseline TMD–STA relationship (gray lines in *Figure 10a* and *Figure 10b*). This relationship not only dictates how the motor system physically controls the redundant system, but also informs the resulting visual feedback. When visual perturbations are applied to the tip-movement direction or the stick-tilt angle (bold arrows in *Figure 10a*), the motor system tries to eliminate the dissociation of the perturbed state from the original state (thin arrows in *Figure 10a*). It even responds to the end-effector irrelevant perturbation (red and blue arrows in *Figure 10a*), because the error in the end-effector irrelevant dimension implies the presence of an end-effector relevant error (black dashed arrows in *Figure 10a*) under the condition that the stick-tilt angle is tightly coupled with the tip-movement direction (i.e. the baseline TMD–STA relationship). However, these correction patterns are not physically allowed. Since the tip-movement direction and the stick-tilt angle are physically constrained by the baseline TMD–STA relationship, these correction patterns are biased toward the relationship (bold arrows in *Figure 10b*).

In summary, this study shows that the baseline TMD–STA relationship, originally developed to control the redundant system, also plays a crucial role for the motor system in how it should implicitly respond to the visual errors and how it should correct the physical movement. This idea provides a novel framework for understanding how the motor system leans to control the redundant system when encountering visual errors.

## Materials and methods
### Participants
A total of 72 healthy adults (25 females and 47 males, aged 22.1±2.5 years old; mean ± SD) participated in the study. Participants were recruited using an online recruitment system (https://www.jikken-baito.com). All the participants were right-handed according to their laterality score (83.3±13.7; mean ± SD) obtained using the Edinburgh Handedness Inventory (*Oldfield, 1971*). They provided written informed consent.

### Experimental apparatus
The participants performed reaching movements using a virtual stick displayed on a horizontal screen with both handles of a manipulandum (KINARM End-Point Lab; Kinarm, Kingston, Canada; *Scott, 1999*; *Figure 2a*). This screen prevented the participants from directly seeing their arms and handles, and displayed a start position, a target (1.4 cm in diameter), and a virtual stick (length: 40 cm, width: 0.5 cm). A white circle (1 cm diameter) was displayed on the right tip of the stick. The participants held the left tip of the stick with their left hand (left handle) and the stick 15 cm away from the left hand with their right hand (right handle). The distance between the handles was fixed using a strong elastic force produced by the manipulandum (spring constant = 2000 N/m). The positions and velocities of the hands were analogue/digitally converted at 1.129 kHz and then recorded at 1 kHz for offline analysis.

### Baseline phase
All the experiments (Experiments 1–3) comprised a baseline phase (360 trials) followed by an adaptation phase (240 trials). The baseline phase, which was common to all experiments, was performed to investigate the basic strategy of how the participants moved the stick to reach the targets with the right tip. Before each reaching movement, the participants were asked to move the stick to the home position with the help of a manipulandum (*Figure 2c*). After maintaining the home position for 0.5–1 s, a gray target was pseudo-randomly chosen from nine possible targets (0°: horizontal direction,±10°,±20°,±30°,±40°) appearing 10 cm away from the initial right tip position of the stick. After waiting for another 0.5–1 s, the color of the target changed to pink, indicating that they should immediately start moving the tip to the target. The participants were instructed to maintain the peak velocities of the tip as constant as possible across the trials. When the tip reached the target, a feedback message, 'fast' or 'slow', was presented on the monitor if the peak movement speed was higher or lower than the range of 300–450 mm/s, respectively. After maintaining the tip at the target for

0.5 s, the visual feedback of the stick disappeared. Then, the handles automatically returned to their home position, and the next trial began. During the subsequent 240 trials in the adaptation phase, the target appeared only in the 0° direction, and distinct visual perturbations were introduced in Experiments 1–3 as described below. The home positions, size of the visual feedback and the target, inter-trial interval, and criteria for the tip velocity feedback for the adaptation phase were the same as those in the baseline phase.

## Experiment 1

Experiment 1 A (N=19) and Experiment 1B (N=13) were performed to investigate how participants moved the stick when adapting to end-effector relevant perturbations (tip perturbations). In Experiment 1 A, the tip-movement direction on the monitor was gradually rotated CCW direction around the starting position by 1° per trial up to 30° (*Figure 4a*). The stick-tilt angle remained unchanged (i.e., the stick-tilt angle was identical to the angle of the line connecting both hands). In Experiment 1 A, participants did not perceive the presence of rotation and the adaptation was considered to be dominated by the implicit process. On the other hand, in Experiment 1B, the 30° visual rotation was abruptly applied to the tip-movement direction, which allowed them to use an explicit strategy.

## Experiment 2

Experiment 2 (N=20) examined whether the motor system was indifferent to end-effector irrelevant perturbations (stick-tilt perturbations). The stick-tilt angle on the monitor was rotated around the right tip in the CW direction (E2CW group, N=10) or in the CCW direction (E2CCW group, N=10). Notably, these perturbations did not affect the tip-movement direction. During each reaching trial, the stick-tilt angle increased from 0° to 6° in proportion to the distance of the tip position from the starting point (0.6°/cm; *Figure 6a*). Participants did not notice any perturbations during the experiment.

## Experiment 3

In Experiment 3 (N=20), to examine how end-effector irrelevant stick-tilt perturbations influenced adaptation to end-effector relevant tip perturbations, both tip-movement and stick-tilt perturbations were simultaneously imposed (*Figure 8a*). Ten participants participated in a task in which CCW tip and CCW stick-tilt rotations were imposed (E3CCW group). The remaining 10 participants participated in a task in which CCW tip and CW stick-tilt rotations were imposed (E3CW group). The procedure for applying tip and stick-tilt perturbations was the same as in Experiment 1 A and Experiment 2, respectively. The participants did not notice perturbations during the experiment.

## Data analysis

Data analyses were performed using MATLAB R2023a (MathWorks, Natick, Massachusetts, USA). The behavioral data were low-pass filtered (10 Hz, fourth-order Butterworth filter). The tip-movement direction was calculated as the movement direction of the tip relative to the horizontal direction (*Figure 3a*). The stick-tilt angle was defined as the tilting angle of the stick from the horizontal line (*Figure 3a*). Positive values for both parameters were defined as the degree to the CCW direction. Unless otherwise noted, the tip-movement direction and the stick-tilt angle at the tip's peak velocity were used to evaluate the feedforward motor commands.

### Minimum distance of hand movement

The total movement distance of both hands in *Figure 3b* and *Figure 3c* was estimated by summing the Euclidean distances between the initial and final positions of the hands. The distance is represented as a function of the stick-tilt angle at the movement offset ($\theta$).

$$
\begin{aligned}
f(\phi, \theta) \quad &= \sqrt{(d\cos\phi - L\cos\theta + L)^2 + (d\sin\phi - L\sin\theta)^2} \\
&+ \sqrt{(d\cos\phi - R\cos\theta + R)^2 + (d\sin\phi - R\sin\theta)^2}
\end{aligned}
\tag{1}
$$

The first term on the right side represents the distance of the left-hand movement, whereas the second term represents the distance of the right-hand movement. φ, d, L, and R represent a target direction (0°,±10°,±20°,±30°, and ±40°), a target distance (10 cm), a distance between the left hand

and the tip (40 cm), and a distance between the right hand and the tip (25 cm), respectively. The minimum distance of hand movement was derived by solving the following equation:

$$\frac{\partial f}{\partial \theta} = 0 \qquad (2)$$

## Statistical analysis

To statistically evaluate the behavioral data, we conducted $t$-tests, correlation analyses, two-way repeated-measures ANOVA, and MANOVA using MATLAB, the software package JASP (http://jasp-stats.org), and R studio.

## Acknowledgements

We thank members of the Nozaki laboratory for their helpful comments and suggestions, Jinnosuke Ueno, Junnosuke Sakuragi, and Ryosuke Miyazawa for performing preliminary experiments, Asako Munakata and Mayumi Yoda for coordinating experiments. This study was supported by a grant from the Japan Society for the Promotion of Science Research Fellowships for Young Scientists to TK (20J13734) and a KAKENHI to DN (17H00874, 21H04860) and to TK (23K10739).

## Additional information

### Funding

| Funder | Grant reference number | Author |
|---|---|---|
| Japan Society for the Promotion of Science | 20J13734 | Toshiki Kobayashi |
| Japan Society for the Promotion of Science | 23K10739 | Toshiki Kobayashi |
| Japan Society for the Promotion of Science | 17H00874 | Daichi Nozaki |
| Japan Society for the Promotion of Science | 21H04860 | Daichi Nozaki |

The funders had no role in study design, data collection and interpretation, or the decision to submit the work for publication.

### Author contributions

Toshiki Kobayashi, Conceptualization, Formal analysis, Investigation, Methodology, Writing - original draft, Writing - review and editing; Daichi Nozaki, Conceptualization, Formal analysis, Supervision, Investigation, Methodology, Writing - original draft, Writing - review and editing

### Author ORCIDs

Daichi Nozaki [iD] https://orcid.org/0000-0002-1338-8337

### Ethics

Human subjects: The ethics committee of the University of Tokyo approved the experimental protocol (19-225). The participants provided written informed consent.

Reviewer #1 (Public Review): https://doi.org/10.7554/eLife.96665.3.sa1
Reviewer #2 (Public review): https://doi.org/10.7554/eLife.96665.3.sa2
Reviewer #3 (Public review): https://doi.org/10.7554/eLife.96665.3.sa3
Author response https://doi.org/10.7554/eLife.96665.3.sa4

## Additional files

### Supplementary files
• MDAR checklist

### Data availability
Data collected in Experiments 1-3 and the code are uploaded in zenodo and are available at https://zenodo.org/record/8033232.

The following dataset was generated:

| Author(s) | Year | Dataset title | Dataset URL | Database and Identifier |
|---|---|---|---|---|
| Kobayashi T | 2023 | Motor control and adaptation in a redundant motor task manipulating a stick with both hands | https://doi.org/10.5281/zenodo.8033232 | Zenodo, 10.5281/zenodo.8033232 |

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
