## [Editor Report · eLife Assessment]

This study presents a **valuable** finding on how the sensorimotor control system deals with redundancy within our body, based on a novel bimanual task. The evidence supporting the authors' claims is **convincing**, as demonstrated over four different experiments. The work will be of interest to researchers from the motor control community and related fields, and further investigation into the interpretation of the findings could increase the generalisation of the study to a broader audience.

---

## [Referee Report · Reviewer #1 (Public Review)]

Summary/Strengths:

This manuscript describes a stimulating contribution to the field of human motor control. The complexity of control and learning is studied with a new task offering a myriad of possible coordination patterns. Findings are original and exemplify how baseline relationships determine learning.

Weaknesses:

A new task is presented: it is a thoughtful one, but because it is a new one, the manuscript section is filled with relatively new terms and acronyms that are not necessarily easy to rapidly understand.

First, some more thoughts may be devoted to the take-home message. In the title, I am not sure manipulating a stick with both hands is a key piece of information. Also, the authors appear to insist on the term 'implicit', and I wonder if it is a big deal in this manuscript and if all the necessary evidence appears in this study that control and adaptation are exclusively implicit. As there is no clear comparison between gradual and abrupt sessions, the authors may consider removing at least from the title and abstract the words 'implicit' and 'implicitly'. Most importantly, the authors may consider modifying the last sentence of the abstract to clearly provide the most substantial theoretical advance from this study.

It seems that a substantial finding is the 'constraint' imposed by baseline control laws on sensorimotor adaptation. This seems to echo and extend previous work of Wu, Smith et al. (Nat Neurosci, 2014): their findings, which were not necessarily always replicated, suggested that the more participants were variable in baseline, the better they adapted to a systematic perturbation. The authors may study whether residual errors are smaller or adaptation is faster for individuals with larger motor variability in baseline. Unfortunately, the authors do not present the classic time course of sensorimotor adaptation in any experiment. The adaptation is not described as typically done: the authors should thus show the changes in tip movement direction and stick-tilt angle across trials, and highlight any significant difference between baseline, early adaptation, and late adaptation, for instance. I also wonder why the authors did not include a few no-perturbation trials after the exposure phase to study after-effects in the study design: it looks like a missed opportunity here. Overall, I think that showing the time course of adaptation is necessary for the present study to provide a more comprehensive understanding of that new task, and to re-explore the role of motor variability during baseline for sensorimotor adaptation.

The distance between hands was fixed at 15 cm with the Kinarm instead of a mechanical constraint. I wonder how much this distance varied and more importantly whether from that analysis or a force analysis, the authors could determine whether one hand led the other one in the adaptation.

I understand the distinction between task- and end-effector irrelevant perturbation, and at the same time results show that the nervous system reacts to both types of perturbation, indicating that they both seem relevant or important. In line 32, the errors mentioned at the end of the sentence suggest that adaptation is in fact maladaptive. I think the authors may extend the Discussion on why adaptation was found in the experiments with end-effector irrelevant and especially how an internal (forward) model or a pair of internal (forward) models may be used to predict both the visual and the somatosensory consequences of the motor commands.

---

## [Referee Report · Reviewer #2 (Public review)]

Summary:

The authors have developed a novel bimanual task that allows them to study how the sensorimotor control system deals with redundancy within our body. Specifically, the two hands control two robot handles that control the position and orientation of a virtual stick, where the end of the stick is moved into a target. This task has infinite solutions to any movement, where the two hands influence both tip-movement direction and stick-tilt angle. When moving to different targets in the baseline phase, participants change the tilt angle of the stick in a specific pattern that produces close to minimum movement of the two hands to produce the task. In a series of experiments, the authors then apply perturbations to the stick angle and stick movement direction to examine how either tip-movement (task-relevant) or stick-angle (task-irrelevant) perturbations effect adaptation. Both types of perturbations affect adaptation, but this adaptation follows the baseline pattern of tip-movement and stick angle relation such that even task-irrelevant perturbations drive adaptation in a manner that results in task-relevant errors. Overall, the authors suggest that these baseline relations affect how we adapt to changes in our tasks. This work provides an important demonstration that underlying solutions\relations can affect the manner in which we adapt. I think one major contribution of this work will also be the task itself, which provides a very fruitful and important framework for studying more complex motor control tasks.

Strengths:

Overall, I find this a very interesting and well-written paper. Beyond providing a new motor task that could be influential in the field, I think it also contributes to studying a very important question - how we can solve redundancy in the sensorimotor control system, as there are many possible mechanisms or methods that could be used - each of which produces different solutions and might affect the manner in which we adapt.

Weaknesses:

The visual perturbations were only provided while reaching to one target, which limits the amount of exploration of the environment that the participants experience. Overall, I would find the results even more compelling if the same perturbations applied to movements to more (or all) of the targets produced similar adaptation profiles. The question is to what degree the results derive from only providing a small subset of the environment to explore.

---

## [Referee Report · Reviewer #3 (Public review)]

Summary:

This study investigated motor system adaptation to new environments through modifications in redundant body movements. Utilizing a novel bimanual stick-manipulation task, participants controlled a virtual stick to reach targets, focusing on how tip-movement direction perturbations affected tip movement and stick-tilt adaptation. The findings revealed a consistent strategy among participants who flexibly adjusted the tilt angle of the stick in response to errors. The adaptation patterns were influenced by physical space relationships, which guided the motor system's selection of movement patterns. This study underscores the motor system's adaptability through changes in redundant body movement patterns.

Strengths:

This study introduces an innovative bimanual stick manipulation task to explore motor system adaptation to novel environments through alterations in redundant body movement patterns. It also expands the use of endpoint robots in motor control studies.

Weaknesses:

The generalizability of the findings is limited. Future work may strengthen the present study's findings by examining whether the observed relationships hold for different stick lengths (i.e., varying hand positions along the virtual stick) or when reaching targets to the left and right of the starting position, not just at varying angles along one side. Additionally, a more comprehensive review of the existing literature on redundant systems, rather than primarily focusing on the lack of redundancy in endpoint-reaching tasks, would have strengthened this study. While the novel task expands the use of endpoint robots in motor control studies, its utility in exploring broader aspects of motor control and learning may be constrained.

---

## [Author Response]

The following is the authors’ response to the original reviews.

We would like to thank all the reviewers for their positive evaluation of our paper, as described in the Strengths section. We are also grateful for their helpful comments and suggestions, which we have addressed below. We believe that the manuscript has been significantly improved as a result of these suggestions. In addition to these changes, we also corrected some inconsistencies (statistical values in the last sentence of a Figure 5 caption) and sentences in the main text (lines 155, 452, 522) (these corrections did not affect the results).

Fig. 5e: R=0.599, P<0.001 -> R=0.601, P=0.007

L150: "the angle of stick tilt angle" -> "the angle of stick tilt"

L437: "no such" -> "such"

L522: "?" -> "."

**Reviewer #1 (Public Review):**
Summary/Strengths:This manuscript describes a stimulating contribution to the field of human motor control. The complexity of control and learning is studied with a new task offering a myriad of possible coordination patterns. Findings are original and exemplify how baseline relationships determine learning.Weaknesses:A new task is presented: it is a thoughtful one, but because it is a new one, the manuscript section is filled with relatively new terms and acronyms that are not necessarily easy to rapidly understand.First, some more thoughts may be devoted to the take-home message. In the title, I am not sure manipulating a stick with both hands is a key piece of information. Also, the authors appear to insist on the term ‘implicit’, and I wonder if it is a big deal in this manuscript and if all the necessary evidence appears in this study that control and adaptation are exclusively implicit. As there is no clear comparison between gradual and abrupt sessions, the authors may consider removing at least from the title and abstract the words ‘implicit’ and ‘implicitly’. Most importantly, the authors may consider modifying the last sentence of the abstract to clearly provide the most substantial theoretical advance from this study.

Thank you for your positive comment on our paper. We agree with the reviewer that our paper used a lot of acronyms that might confuse the readers. As we have addressed below (in the rebuttal to the Results section), we have reduced the number of acronyms.

Regarding the comment on the use of the word “implicit” in the title and the abstract, we believe that its use in this paper is very important and indispensable. One of our main findings was that the pattern of adaptation between the tip-movement direction and the stick-tilt angle largely followed that in the baseline condition when aiming at different target directions. This adaptation was largely implicit because participants were not aware of the presence of the perturbation as the amount of perturbation was gradually increased. This implicitness suggests that the adaptation pattern of how the movement should be corrected is embedded in the motor learning system. On the other hand, if this adaptation pattern was achieved on the basis of the explicit strategy of changing the direction of the tip-movement, the adaptation pattern that follows the baseline pattern is not at all surprising. For these reasons, we will continue to use the word "implicit".

It seems that a substantial finding is the ‘constraint’ imposed by baseline control laws on sensorimotor adaptation. This seems to echo and extend previous work of Wu, Smith et al. (Nat Neurosci, 2014): their findings, which were not necessarily always replicated, suggested that the more participants were variable in baseline, the better they adapted to a systematic perturbation. The authors may study whether residual errors are smaller or adaptation is faster for individuals with larger motor variability in baseline. Unfortunately, the authors do not present the classic time course of sensorimotor adaptation in any experiment. The adaptation is not described as typically done: the authors should thus show the changes in tip movement direction and stick-tilt angle across trials, and highlight any significant difference between baseline, early adaptation, and late adaptation, for instance. I also wonder why the authors did not include a few noperturbation trials after the exposure phase to study after-effects in the study design: it looks like a missed opportunity here. Overall, I think that showing the time course of adaptation is necessary for the present study to provide a more comprehensive understanding of that new task, and to re-explore the role of motor variability during baseline for sensorimotor adaptation.

We appreciate the reviewer for raising these important issues.

Regarding the learning curve, because the amount of perturbation was gradually increased except for Exp.1B, we were not able to obtain typical learning curves (i.e., the curve showing errors decaying exponentially with trials). However, it may still be useful to show how the movement changed with trials during adaptation. Therefore, following the reviewer's suggestion, we have added the figures of the time course of adaptation in the supplementary data (Figures S1, S2, S4, and S5).

There are two reasons why our experiments did not include aftereffect quantification trials (i.e., probe trials). First, in the case of adaptation to a visual perturbation (e.g., visual rotation), probe trials are not necessary because the degree of adaptation can be easily quantified by the amount of compensation in the perturbation trials (however, in the case of dynamic perturbations such as force fields, the use of probe trials is necessary). Second, the inclusion of probe trials allows participants to be aware of the presence of the perturbation, which we would like to avoid.

We also appreciate the interesting additional questions regarding the relevance of our work to the relationship between baseline motor variability and adaptation performance. As this topic, although interesting, is outside the scope of this paper, we concluded that we would not address it in the manuscript. In fact, the experiments were not ideal for quantifying motor variability in the baseline phase because participants had to aim at different targets, which could change the characteristics of motor variability. In addition, we gradually increased the size of the perturbation except for Exp.1B (see Author response image 1, upper panel), which could make it difficult to assess the speed of adaptation. Nevertheless, we think it is worth mentioning this point in this rebuttal. Specifically, we examined the correlation between baseline motor variability when aiming the 0 deg target (tip-movement direction or stick-tilt angle) and adaptation speed in Exp 1A and Exp 1B (Author response image 1 and Author response image 2). To assess adaptation speed in Exp.1A, we quantified the slope of the tip-movement direction to a gradually increasing perturbation (Author response image 1, upper panel). The adaptation speed in Exp.1B was obtained by fitting the exponential function to the data (Author response image 2, upper panel). Although the statistical results were not completely consistent, we found that the participants with greater the motor variability at baseline tended to show faster adaptation, as shown in a previous study (Wu et al., Nat Neurosci, 2014).

**Author response image 1. sa4fig1:** Correlation between the baseline variability and learning speed (Experiment 1A). In Exp 1A, the rotation of the tip-movement direction was gradually increased by 1 degree per trial up to 30 degrees. The learning speed was quantified by calculating how quickly the direction of movement followed the perturbation (upper panel). The lower left panel shows the variability of the tip-movement direction versus learning speed, while the lower right panel shows the variability of the stick-tilt angle versus learning speed. Baseline variability was calculated as a standard deviation across trials (trials in which a target appeared in a 0-degree direction).

**Author response image 2. sa4fig2:** Correlation between the baseline variability and learning speed (Experiment 1B). In Exp 1B, the rotation of the tip-movement direction was abruptly applied from the first trial (30 degrees). The learning speed was calculated as a time constant obtained by exponential curve fitting. The lower left panel shows the variability of the tip-movement direction versus learning speed, while the lower right panel shows the variability of the stick-tilt angle versus learning speed. Baseline variability was calculated as a standard deviation across trials (trials in which a target appeared in a 0-degree direction).

The distance between hands was fixed at 15 cm with the Kinarm instead of a mechanical constraint. I wonder how much this distance varied and more importantly whether from that analysis or a force analysis, the authors could determine whether one hand led the other one in the adaptation.

Thank you very much for this important comment. Since the distance between the two hands was maintained by the stiff virtual spring (2000 N/m), it was kept almost constant throughout the experiments as shown in Author response image 3 (the averaged distance during a movement). The distance was also maintained during reaching movements (Author response image 4).

We also thank the reviewer for the suggestion regarding the force analysis. As shown in Author response image 5, we did not find a role for a specific hand for motor adaptation from the handle force data. Specifically, Author response image 5 shows the force applied to each handle along and orthogonal to the stick. If one hand led the other in adaptation, we should have observed a phase shift as adaptation progressed. However, no such hand specific phase shift was observed. It should be noted, however, that it was theoretically difficult to know from the force sensors which hand produced the force first, because the force exerted by the right handle was transmitted to the left handle and vice versa due to the connection by the stiff spring.

**Author response image 3. sa4fig3:** The distance between hands during the task. We show the average distance between hands for each trial. The shaded area indicates the standard deviation across participants.

**Author response image 4. sa4fig4:** Time course changes in the distance between hands during the movement. The color means the trial epoch shown in the right legend.

**Author response image 5. sa4fig5:** The force profile during the movement (Exp 1A). We decomposed the force of each handle into the component along (upper panels) and orthogonal to the stick (lower panels). Changes in the force profiles in the adaptation phase are shown (*left*: left hand force, *right*: right hand force). The colors (magenta to cyan) mean trial epoch shown in the right legend.

I understand the distinction between task- and end-effector irrelevant perturbation, and at the same time results show that the nervous system reacts to both types of perturbation, indicating that they both seem relevant or important. In line 32, the errors mentioned at the end of the sentence suggest that adaptation is in fact maladaptive. I think the authors may extend the Discussion on why adaptation was found in the experiments with end-effector irrelevant and especially how an internal (forward) model or a pair of internal (forward) models may be used to predict both the visual and the somatosensory consequences of the motor commands.

Thank you very much for your comment. As we already described in the discussion of the original manuscript (Lines 519-538 in the revised manuscript), two potential explanations exist for the motor system’s response to the end-effector irrelevant perturbation (i.e., stick rotation). First, the motor system predicts the sensory information associated with the action and attempts to correct any discrepancies between the prediction and the actual sensory consequences, regardless of whether the error information is end-effector relevant or end-effector irrelevant. Second, given the close coupling between the tip-movement direction and stick-tilt angle, the motor system can estimate the presence of end-effector relevant error (i.e., tip-movement direction) by the presence of end-effector irrelevant error (i.e., stick-tilt angle). This estimation should lead to the change in the tip-movement direction. As the reviewer pointed out, the mismatch between visual and proprioceptive information is another possibility, we have added the description of this point in Discussion (Lines 523-526).

**Reviewer #1 (Recommendations For The Authors):**
MinorLine 16: “it remains poorly understood” is quite subjective and I would suggest reformulating this statement.

We have reformulated this statement as “This limitation prevents the study of how….” (Line 16).

IntroductionLine 49: the authors may be more specific than just saying ‘this task’. In particular, they need to clarify that there is no redundancy in studies where the shoulder is fixed and all movement is limited to a plane ... which turns out to truly happen in a limited set of experimental setups (for example: Kinarm exoskeleton, but not endpoint; Kinereach system...).

We have changed this to “such a planar arm-reaching task” (Line 49).

Line 61: large, not infinite because of biomechanical constraints.

We have changed “an infinite” to “a large” (Line 61) and “infinite” to “a large number of” (legend in Fig. 1f).

Lines 67-69: consider clarifying.

We have tried to clarify the sentence (Lines 67-69).

ResultsTMD and STA, and TMD-STA plane, are new terms with new acronyms that are not easy to immediately understand. Consider avoiding acronyms.

We have reduced the use of these acronyms as much as possible.

“visual TMD–STA plane” -> “plane representing visual movement patterns” (Lines 179180)

“TMD axis” -> “x-axis” (Line 181, Line 190)

“physical TMD–STA plane” -> “plane representing physical movement patterns” (Lines 182-187)

“physical TMD–STA plane” -> “physical plane” (Line 191, Line 201, Lines 216-217, Line 254, Line 301, Line 315, Line 422, Line 511, and captions of Figures 4-9, S3)

“visual TMD–STA plane” -> “visual plane” (Line 193, Line 241, Line 248, Line 300, Lines 313-314, and captions of Figures 4-9, S3)

“STA axis” -> “y-axis” (Line 241)

Line 169: please clarify the mismatch(es) that are created when the tip-movement direction is visually rotated in the CCW direction around the starting position (tip perturbation), whereas the stick-tilt angle remains unchanged.

Thank you for your pointing this out. We have clarified that the stick-tilt angle remains identical to the tilt of both hands (Lines 171-172).

DiscussionI understand the physical constraint imposed between the 2 hands with the robotic device, but I am not sure I understand the physical constraint imposed by the TMD-STA relationship.

The phrase “physical constraint” meant the constraint of the movement on the physical space. However, as the reviewer pointed out, this phrase could confuse the constraint between the two hands. Therefore, we have avoided using the phrase “physical constraint” throughout the manuscript.

Some work looking at 3-D movements should be used for Discussion (e.g. Lacquaniti & Soechting 1982; work by d’Avella A or Jarrasse N).

Thank you for sharing this important information. We have cited these studies in Discussion (Lines 380-382).

**Reviewer #2 (Public Review):**
Summary:The authors have developed a novel bimanual task that allows them to study how the sensorimotor control system deals with redundancy within our body. Specifically, the two hands control two robot handles that control the position and orientation of a virtual stick, where the end of the stick is moved into a target. This task has infinite solutions to any movement, where the two hands influence both tip-movement direction and stick-tilt angle. When moving to different targets in the baseline phase, participants change the tilt angle of the stick in a specific pattern that produces close to the minimum movement of the two hands to produce the task. In a series of experiments, the authors then apply perturbations to the stick angle and stick movement direction to examine how either tipmovement (task-relevant) or stick-angle (task-irrelevant) perturbations affect adaptation. Both types of perturbations affect adaptation, but this adaptation follows the baseline pattern of tip-movement and stick angle relation such that even task-irrelevant perturbations drive adaptation in a manner that results in task-relevant errors. Overall, the authors suggest that these baseline relations affect how we adapt to changes in our tasks. This work provides an important demonstration that underlying solutions/relations can affect the manner in which we adapt. I think one major contribution of this work will also be the task itself, which provides a very fruitful and important framework for studying more complex motor control tasks.Strengths:Overall, I find this a very interesting and well-written paper. Beyond providing a new motor task that could be influential in the field, I think it also contributes to studying a very important question - how we can solve redundancy in the sensorimotor control system, as there are many possible mechanisms or methods that could be used - each of which produces different solutions and might affect the manner in which we adapt.Weaknesses:I would like to see further discussion of what the particular chosen solution implies in terms of optimality.The underlying baseline strategy used by the participants appears to match the path of minimum movement of the two hands. This suggests that participants are simultaneously optimizing accuracy and minimizing some metabolic cost or effort to solve the redundancy problem. However, once the perturbations are applied, participants still use this strategy for driving adaptation. I assume that this means that the solution that participants end up with after adaptation actually produces larger movements of the two hands than required. That is - they no longer fall onto the minimum hand movement strategy - which was used to solve the problem. Can the authors demonstrate that this is either the case or not clearly? These two possibilities produce very different implications in terms of the results.If my interpretation is correct, such a result (using a previously found solution that no longer is optimal) reminds me of the work of Selinger et al., 2015 (Current Biology), where participants continue to walk at a non-optimal speed after perturbations unless they get trained on multiple conditions to learn the new landscape of solutions. Perhaps the authors could discuss their work within this kind of interpretation. Do the authors predict that this relation would change with extensive practice either within the current conditions or with further exploration of the new task landscape? For example, if more than one target was used in the adaptation phase of the experiment?On the other hand, if the adaptation follows the solution of minimum hand movement and therefore potentially effort, this provides a completely different interpretation.Overall, I would find the results even more compelling if the same perturbations applied to movements to all of the targets and produced similar adaptation profiles. The question is to what degree the results derive from only providing a small subset of the environment to explore.

Thank you very much for pointing out this significant issue. As the reviewer correctly interprets, the physical movement patterns deviated from the baseline relationship as exemplified in Exp.2. However, this deviation is not surprising for the following reason. Under the perturbation that creates the dissociation between the hands and the stick, the motor system cannot simultaneously return both the visual stick motion and physical hands motion to the original motions: When the motor system tries to return the visual stick motion to the original visual motion, then the physical hands motion inevitably deviates from the original physical hands motion, and vice versa.

Our interpretation of this result is that the motor system corrects the movement to reduce the visual dissociation of the visual stick motion from the baseline motion (i.e., sensory prediction error), but this movement correction is biased by the baseline physical hands motion. In other words, the motor system attempts to balance the minimization of sensory prediction error and the minimization of motor cost. Thus, our results do not indicate that the final adaptation pattern is non-optimal, but rather reflect the attempts for optimization.

In the revised manuscript, we have added the description of this interpretation (Lines 515-517).

**Reviewer #2 (Recommendations For The Authors):**
The authors have suggested that the only study (line 472) that has also examined an end-effector irrelevant perturbation is the bimanual study of Omrani et al., 2013, which only examined reflex activity rather than adaptation. To clarify this issue - exactly what is considered end-effector irrelevant perturbations - I was wondering about the bimanual perturbations in Dimitriou et al., 2012 (J Neurophysiol) and the simultaneous equal perturbations in Franklin et al., 2016 (J Neurosci), as well as other recent papers studying task-irrelevant disturbances which aren’t discussed. I would consider these both to also be end-effector irrelevant perturbations, although again they only used these to study reflex activity and not adaptation as in the current paper. Regardless, further explanation of exactly what is the difference between task-irrelevant and end-effector irrelevant would be useful to clarify the exact difference between the current manuscript and previous work.

Thank you for your helpful comments. We have included as references the study by Dimitriou et al. (Line 490) and Franklin et al. (Lines 486-487), which use an endeffector irrelevant perturbation and the task-irrelevant perturbation condition, respectively. We have also added further explanation of what is the difference between task-irrelevant and end-effector irrelevant (Lines 344-352).

Line 575: I assume that you mean peak movement speed

We have added “peak”. (Line 597).

**Reviewer #3 (Public Review):**
Summary:This study explored how the motor system adapts to new environments by modifying redundant body movements. Using a novel bimanual stick manipulation task, participants manipulated a virtual stick to reach targets, focusing on how tip-movement direction perturbations affected both tip movement and stick-tilt adaptation. The findings indicated a consistent strategy among participants who flexibly adjusted the tilt angle of the stick in response to errors. The adaptation patterns are influenced by physical space relationships, guiding the motor system’s choice of movement patterns. Overall, this study highlights the adaptability of the motor system through changes in redundant body movement patterns.Strengths:This paper introduces a novel bimanual stick manipulation task to investigate how the motor system adapts to novel environments by altering the movement patterns of our redundant body.Weaknesses:The generalizability of the findings is quite limited. It would have been interesting to see if the same relationships were held for different stick lengths (i.e., the hands positioned at different start locations along the virtual stick) or when reaching targets to the left and right of a start position, not just at varying angles along one side. Alternatively, this study would have benefited from a more thorough investigation of the existing literature on redundant systems instead of primarily focusing on the lack of redundancy in endpointreaching tasks. Although the novel task expands the use of endpoint robots in motor control studies, the utility of this task for exploring motor control and learning may be limited.

Thank you very much for the important comment. Given that there are many parameters (e.g., stick length, locations of hands, target position etc), one may wonder how the findings obtained from only one combination can be generalized to other configurations. In the revised manuscript, we have explicitly described this point (Lines 356-359).

Thus, the generalizability needs to be investigated in future studies, but we believe that the main results also apply to other configurations. Regarding the baseline stick movement pattern, the control with tilting the stick was observed regardless of the stick-tip positions (Author response image 6). Regarding the finding that the adapted stick movement patterns follow the baseline movement patterns, we confirmed the same results even when the other targets were used as the target for the adaptation (Author response image 7).

**Author response image 6. sa4fig6:** Stick-tip manipulation patterns when the length of the stick varied.

**Author response image 7. sa4fig7:** Patterns of adaptation when using the other targets. In the baseline phase, 40 naïve participants moved a stick tip to a peripheral target (24 directions). They showed a stereotypical relationship between the tip-movement direction and the stick-tilt angle (a bold gray curve). In the adaptation phase, participants were divided into four groups, each with a different target training direction (lower left, lower right, upper right, or upper left), and visual rotation was gradually imposed on the tip-movement direction. Irrespective of the target direction, the adaptation pattern of the tipmovement and stick-tilt followed with the baseline relationship.

We also thank you for your comment about studying the existing redundant systems. We can understand the reviewer's concern about the usefulness of our task, but we believe that we have proposed the novel framework for motor adaptation in the redundant system. The future studies will be able to clarify how the knowledge gained from our task can be generally applied to understand the control and learning of the redundant system.

**Reviewer #3 (Recommendations For The Authors):**
Line 49: replace “uniquely” with primarily. A number of features of the task setup could affect the joint angles, from if/how the arm is supported, whether the wrist is fixed, alignment of the target in relation to the midline of the participant, duration of the task, and whether fatigue is an issue, etc. Your statement relates to fixed limb lengths of a participant, rather than standard reaching tasks as a whole. Not to mention the degree of inter- and intra-subject variability that does exist in point-to-point reaching tasks.

Thank you for your helpful point. We have replaced “uniquely” with “primarily”. (Line 49).

Line 72: the cursor is not an end-effector - it represents the end-effector.

We have changed the expression as “the perturbation to the cursor representing the position of the end-effector (Line 72).

Lines 73 – 78: it would benefit the authors to consider the role of intersegmental dynamics.

Thank you for your suggestion. We are not sure if we understand this suggestion correctly, but we interpret that this suggestion to mean that the end-effector perturbation can be implemented by using the perturbation that considers the intersegmental dynamics. However, the implementation is not so straightforward, and the panels in Figure 1j,k are only conceptual for the end-effector irrelevant perturbation. Therefore, we have not described the contribution of intersegmental dynamics here.

Lines 90 – 92: “cannot” should be “did not”, as the studies being referenced are already completed. This statement should be further unpacked to explain what they did do, and how that does not meet the requirement of redundancy in movement patterns.

We have changed “cannot” to “did not” (Line 91). We have also added the description of what the previous studies had demonstrated (Line 88-90).

Figure text could be enlarged for easier viewing.

We have enlarged texts in all figures.

Lines 41 - 47: Interesting selection of supporting references. For the introduction of a novel environment, I would recommend adding the support of Shadmehr and MussaIvaldi 1994.

Thank you for your suggestion. We have added Shadmehr and Mussa-Ivaldi 1994 as a reference (Line 45).

Line 49: “this task” is vague - the above references relate to a number of different tasks. For example, the authors could replace it with a reaching task involving an end-point robot.

Thank you very much for your suggestion. As per the suggestion by Reviewer #1, we have changed this to “such a planar arm-reaching task” (Line 49).

Line 60: “hypothetical limb with three joints” - in Figure 1a, the human subject, holding the handle of a robotic manipulandum does have flexibility around the wrist.

Previous studies using planar arm-reaching task have constrained the wrist joint (e.g., Flash & Hogan, 1985; Gordon et al., 1994; Nozaki et al., 2006). We tried to emphasize this point as “participants manipulate a visual cursor with their hands *primarily* by moving their shoulder and elbow joints” (Line 42). In the revised manuscript, we have also emphasized this point in the legend of Figure 1a.

Lines 93-108: this paragraph could be cleaned up more clearly stating that while the use of task-irrelevant perturbations has been used in the domain of reaching tasks, the focus of these tasks has not been specifically to address “In our task, we aim to exploit this feature by doing”

Thank you very much for your helpful comments. To make this paragraph clear, we have modified some sentences (Line 100-104).

Line 109: “coordinates to adapt” is redundant.

We have changed this to “adapts” (Line 110).

Lines 109-112: these sentences could be combined to have better flow.

Thank you very much for your valuable suggestion. We have combined these two sentences for the better flow (Line 110-112).

Line 113-114: consider rewording - “This is a redundant task because ...” to something like “Redundancy in the task is achieved by acknowledging that ....“.

We have changed the expression according to the reviewer’s suggestion (Line 114).

Line 118: Consider changing “changes” to “makes use of”.

We have changed the expression (Line 119).

Lines 346 - 348: grammar and clarity - “This redundant motor task enables the investigation of adaptation patterns in the redundant system following the introduction of perturbations that are either end-effector relevant, end-effector irrelevant, or both.“.

Thank you very much again for your helpful suggestion of English expression. We have adopted the sentence you suggested (Line 354-356).